# Clear-Sky Ultraviolet radiation modelling using output from the Chemistry Climate Model Initiative

Kévin Lamy[1], Thierry Portafaix[1], Béatrice Josse[2], Colette Brogniez[3], Sophie Godin-Beekmann[4], Hassan Bencherif[1,5], Laura Revell[6,7,8], Hideharu Akiyoshi[9], Slimane Bekki[4], Michaela I. Hegglin[10], Patrick Jöckel[11], Oliver Kirner[12], Ben Liley[13], Virginie Marecal[2], Olaf Morgenstern[13], Andrea Stenke[6], Guang Zeng[13], N. Luke Abraham[14,15], Alexander T. Archibald[14], Neil Butchart[16], Martyn P. Chipperfield[17], Glauco Di Genova[18], Makoto Deushi[19], Sandip S. Dhomse[17], Rong-Ming Hu[4], Douglas Kinnison[20], Michael Kotkamp[13], Richard McKenzie[13], Martine Michou[2], Fiona M. O'Connor[16], Luke D. Oman[21], Giovanni Pitari[18], David A. Plummer[22], John A. Pyle[14], Eugene Rozanov[6,23], David Saint-Martin[2], Kengo Sudo[24], Taichu Y. Tanaka[19], Daniele Visioni[25], and Kohei Yoshida[19]

[1]LACy, Laboratoire de l'Atmosphère et des Cyclones (UMR 8105 CNRS, Université de La Réunion, Météo-France), Saint-Denis de La Réunion, France
[2]Centre National de Recherches Météorologiques (CNRM) UMR 3589, Météo-France/CNRS, Toulouse, France
[3]Laboratoire d'Optique Atmosphérique (LOA), Université de Lille, Faculté des Sciences et Technologies, Villeneuve d'Ascq, France
[4]Laboratoire Atmosphères, Milieux, Observations Spatiales, Service d'Aéronomie (LATMOS), CNRS, Institut Pierre Simon Laplace, Pierre et Marie Curie University, Paris, France
[5]School of Chemistry and Physics, University of KwaZulu Natal, Durban, South Africa
[6]Institute for Atmospheric and Climate Science, ETH Zürich (ETHZ), Zürich, Switzerland
[7]Bodeker Scientific, Christchurch, New Zealand
[8]School of Physical and Chemical Sciences, University of Canterbury, Christchurch, New Zealand
[9]National Institute of Environmental Studies (NIES), Tsukuba, Japan
[10]Department of Meteorology, University of Reading, Reading, UK
[11]Institut für Physik der Atmosphäre, Deutsches Zentrum für Luft- und Raumfahrt (DLR), Oberpfaffenhofen, Germany
[12]Steinbuch Centre for Computing, Karlsruhe Institute of Technology, Karlsruhe, Germany
[13]National Institute of Water and Atmospheric Research (NIWA), Wellington, New Zealand
[14]Department of Chemistry, University of Cambridge, Cambridge, UK
[15]National Centre for Atmospheric Science, U.K.
[16]Met Office Hadley Centre (MOHC), Exeter, UK
[17]School of Earth and Environment, University of Leeds, Leeds, UK
[18]Department of Physical and Chemical Sciences, Universitá dell'Aquila, L'Aquila, Italy
[19]Meteorological Research Institute (MRI), Tsukuba, Japan
[20]National Center for Atmospheric Research (NCAR), Boulder, Colorado, USA
[21]National Aeronautics and Space Administration Goddard Space Flight Center (NASA GSFC), Greenbelt, Maryland, USA
[22]Environment and Climate Change Canada, Montréal, Canada
[23]Physikalisch-Meteorologisches Observatorium Davos World Radiation Centre, Davos Dorf, Switzerland
[24]Graduate School of Environmental Studies, Nagoya University, Nagoya, Japan
[25]Sibley School of Mechanical and Aerospace Engineering, Cornell University, Ithaca, NY, USA

**Correspondence:** K. Lamy (kevin.lamy@univ-reunion.fr)

**Abstract.**

We have derived values of the Ultraviolet Index (UVI) at solar noon using the Tropospheric Ultraviolet Model (TUV) driven by ozone, temperature and aerosol fields from climate simulations of the first phase of the Chemistry-Climate Model Initiative (CCMI-1). Since clouds remain one of the largest uncertainties in climate projections, we simulated only the clear-sky UVI. We compared the modelled UVI climatologies against present-day climatological values of UVI derived from both satellite data (the OMI-Aura OMUVBd product) and ground-based measurements (from the NDACC network). Depending on the region, relative differences between the UVI obtained from CCMI/TUV calculations and the ground-based measurements ranged between -5.9% and 10.6%.

We then calculated the UVI evolution throughout the 21st century for the four Representative Concentration Pathways (RCPs 2.6, 4.5, 6.0 and 8.5). Compared to 1960s values, we found an average increase in the UVI in 2100 (of 2-4%) in the tropical belt (30°N-30°S). For the mid-latitudes, we observed a 1.8 to 3.4 % increase in the Southern Hemisphere for RCP 2.6, 4.5 and 6.0, and found a 2.3% decrease in RCP 8.5. Higher increases in UVI are projected in the Northern Hemisphere except for RCP 8.5. At high latitudes, ozone recovery is well identified and induces a complete return of mean UVI levels to 1960 values for RCP 8.5 in the Southern Hemisphere. In the Northern Hemisphere, UVI levels in 2100 are higher by 0.5 to 5.5% for RCP 2.6, 4.5 and 6.0 and they are lower by 7.9% for RCP 8.5.

We analysed the impacts of greenhouse gases (GHGs) and ozone-depleting substances (ODSs) on UVI from 1960 by comparing CCMI sensitivity simulations (1960-2100) with fixed GHGs or ODSs at their respective 1960 levels. As expected with ODS fixed at their 1960 levels, there is no large decrease in ozone levels and consequently no sudden increase in UVI levels. With fixed GHG, we observed a delayed return of ozone to 1960 values, with a corresponding pattern of change observed on UVI, and looking at the UVI difference between 2090s values and 1960s values, we found an 8 % increase in the tropical belt during the summer of each hemisphere.

Finally we show that, while in the Southern Hemisphere the UVI is mainly driven by total ozone column, in the Northern Hemisphere both total ozone column and aerosol optical depth drive UVI levels, with aerosol optical depth having twice as much influence on the UVI as total ozone column does.

# 1 Introduction

After the implementation of the Montreal Protocol, emissions of chlorine and bromine-containing ozone depleting substances (ODSs) have started to decrease and the stratospheric ozone layer is showing signs of recovery (Morgenstern et al., 2008; Solomon et al., 2016). Nonetheless, greenhouse gas (GHG) emissions are generally still increasing and are expected to affect future ozone levels (Fleming et al., 2011; Revell et al., 2012). Global circulation model simulations project that the Brewer Dobson circulation will accelerate over the next century (Butchart, 2014), which would lead to a decrease of ozone levels in the tropics and an enhancement at higher latitudes (Hegglin and Shepherd, 2009). Ozone is one of the major factors affecting surface ultraviolet (UV) radiation.

Exposure to UV radiation has both adverse and beneficial effects on human health. Overexposure increases the risk of skin cancers, e.g. cutaneous malignant melanoma and keratinocyte cancers, and a range of eye diseases. Underexposure increases the risk of vitamin D deficiency; vitamin D is critical to healthy bones. It is common in health research and public health communication to use the UV Index (UVI) (Mc Kinlay and Diffey, 1987) as a measure of erythemally (sunburn) weighted UV irradiance. UV radiation also impacts the biosphere (Erickson III et al., 2015) including aquatic ecosystems, which play a central part in biogeochemical cycles (Hader et al., 2007). Phytoplankton productivity is strongly affected by UV radiation (Smith and Cullen, 1995), which can result in either a positive or negative feedback on climate (Zepp et al., 2007).

The implementation of the Montreal Protocol on ozone depleting substances (ODSs) imposed reductions on emissions of man-made substances that increase stratospheric chlorine and bromine levels (i.e., halocarbons), thereby alleviating concerns about increases in future surface UV radiation (Morgenstern et al., 2008). While this protocol and its amendments drastically reduced the emissions of ODSs, recent studies on the evolution of ozone in a changing climate (Butchart, 2014) raised questions about future surface levels UV radiation (Hegglin and Shepherd, 2009; Bais et al., 2011; Correa et al., 2013; Bais et al., 2015).

Numerous chemistry-climate model (CCM) studies found an acceleration of the Brewer-Dobson circulation (BDC) (Butchart, 2014) due to the increase in atmospheric GHG concentrations. The BDC circulation was proposed by Brewer (1949) and Dobson (1956) to explain the latitudinal distribution of ozone and the amount of water vapor in the stratosphere. The BDC corresponds to a meridional transport in the stratosphere, with ascending air in the tropics and subsidence in the polar latitudes. The mechanism that drives this circulation is the dissipation of Rossby and gravity waves (Holton et al., 1995). Therefore, the strength of the BDC depends on the propagation and breaking of planetary waves. Rind et al. (1990) found that a doubling of carbon dioxide ($CO_2$) would lead to an increase in the residual-mean circulation due to the response from planetary waves, where the residual-mean circulation (Andrews et al., 1987) can be seen as a proxy for the BDC. From the doubled $CO_2$ experiment, Rind et al. (2001) found a 30% increase of the troposphere to stratosphere mass exchange. In addition, an accelerated loss of CFCs will reduce the timescale for ozone to recover (Shepherd, 2008). A strengthening of the BDC and an accelerated recovery of ozone will modify the distribution of ozone in the stratosphere and impact UV radiation at the surface.

While the ozone layer in the stratosphere absorbs UV-B (UV radiation with wavelengths in the range 280-315nm), it is not the only factor affecting surface levels of UV radiation. The distance between the Sun and Earth is responsible for about $\approx 7\%$ of the UV radiation variability on the ground (Frederick et al., 1989). The 11-year solar cycle accounts for about 6%

of the UV radiation variability in the stratosphere (Gray et al., 2010). The solar cycle affects UV radiation through changes in stratospheric ozone, while its direct influence is negligible. Solar zenith angle (SZA) plays a key role for the intensity of surface UV radiation. For larger SZA the path travelled through the atmosphere is longer, hence absorption and scattering increase, affecting the UV radiation response to changes in total ozone column (TOZ) (Brühl and Crutzen, 1989). Clouds and

aerosols also cause variability (Bais et al., 1993). In most cases, clouds attenuate the UV signal at the surface by about 15 to 45% (Calbó et al., 2005). Broken cloud cover can also enhance the surface UV radiation (Lovengreen et al., 2005; Marín et al., 2017). Krzyścin and Puchalski (1998) found a 1.5% increase in erythemal UV for a 10% decrease of aerosol optical depth (AOD) and up to a 30% decreases of UV erythemal can be observed due to biomass burning emissions (Lamy et al., 2018). In the UVA region, a mean reduction of irradiance of 15.2% per unit of AOD slant column has been observed by Kazadzis et al.

(2009). Nitrogen dioxide and sulphur dioxide have also a small effect on UV irradiance (Solomon et al., 1999; Vaida et al., 2003).

In the context of a changing climate and with the use of stratospheric CCM simulations, Hegglin and Shepherd (2009) found a 3.8% increase of UVI in the tropics between the 1960s and 2090s. In the Northern Hemisphere, they found a 9% decrease in the UVI due to increased transport of ozone. As part of the precursor multi-model activity to Chemistry-Climate Model

Initiative (CCMI), Chemistry-Climate Model Validation-2 (CCMVal-2), Bais et al. (2011) also calculated the UVI evolution between 1960 and 2100 and reported a small increase in the tropics of 0.9%, a 7.5% and 9.8% decrease in sorthern and southern high latitudes, and a 4.1% decrease in mid latitudes. In both these studies, the largest UV radiation reduction was found over Antarctica. This is consistent with the expected recovery of the ozone layer.

In these previous studies, UVI were calculated from an analytical formula, which only takes into accounts SZA or TOZ,

or from a radiative transfer model which considers multiple parameters as input. The effect of aerosols changes on the UVI evolution were not considered in those studies.

Following from these studies, we investigate here the evolution of surface UV radiation using the latest simulations from the first phase CCMI. CCMI is a project initiated by Future Earth's IGAC (International Global Atmospheric Chemistry) and the World Climate Research Programme's (WCRP) SPARC (Stratosphere-troposphere Processes and their Role in Climate) as

a successor to and in continuation of the Chemistry-Climate Model Validation Activity (CCMVal) (Eyring et al., 2010) and Atmospheric Chemistry and Climate Model Intercomparison Project (ACCMIP) (Eyring et al., 2013). We use CCMI data and the Tropospheric Ultraviolet Model (TUV) (Madronich et al., 1998) to calculate surface irradiance over the globe. Our present study offers new insights about the role of multiple changing parameters on the UVI evolution along the 21st century. The impact of TOZ, GHGs, ODSs and AOD changes on UVI is analyzed in details for multiple scenarios.

In Section 2, we will explain the methodology used to calculate ground surface irradiance from CCMI data and TUV, and describe the TUV model. We will briefly present the CCMI models along with the different experiments performed for CCMI. A validation of UVI, calculated with CCMI data and TUV, against satellite and ground-based measurements, will be presented in Section 3. A discussion of the spread between CCMI models and on the resulting sensitivity of TUV will also be conducted in this section. In Section 4, we examine the possible evolution of UVI at different latitudinal bands following the representative

concentration pathways (RCPs) (Meinshausen et al., 2011). We also analyse the difference between monthly values of UVI in

the 1960s and 2090s. Sensitivity simulations using concentrations of ODSs and GHGs fixed at constant 1960 levels were also performed for the CCMI exercise. These allow us to assess the impact of the evolution of GHGs and ODSs on UVI individually. An analysis of the impact of AOD on UVI is presented in Section 4.4. The last section will discuss and conclude the findings of the present study.

## 2  Data and methodology

### 2.1  Modelling UV irradiance

UV irradiance at the Earth's surface is calculated with the TUV radiative transfer model (version 5.3) for the entire globe on a 2° by 2° grid. The solar spectral irradiance simulated at the Earth's surface ranges from 280 to 450 nm with a 1 nm step. The solar spectral irradiance is weighted according to the erythemal action spectrum (Mc Kinlay and Diffey, 1987). The resulting weighted solar spectrals irradiance is then integrated from 280 to 450nm to obtain the UVI. For the extra-terrestrial spectrum we used the Dobber et al. (2008) spectrum, for the ozone cross section absorption we used the one from (Gorshelev et al., 2014) and (Serdyuchenko et al., 2014). The required inputs for the UV radiation calculation are:

- Total Nitrogen Dioxide ($TNO_2$)

- Ozone Profile (OP)

- Total Ozone Column (TOZ)

- Temperature Profile (TP)

- Aerosol Optical Depth (AOD)

- Aerosol Ångström exponent ($\alpha$)

- Single Scattering Albedo (SSA)

- Ground Surface Albedo (ALB)

- Altitude (z)

As input for TUV we used data from the latest CCMI simulations (Hegglin and Lamarque, 2015). A brief description of the CTMs or CCMs used in this study is provided in Table 1 and Table 2, while more details are available in Morgenstern et al. (2017). From these models the monthly output from the refC2, senC2rcp26, senC2rcp45, senC2rcp85, senC2fODS and senC2fGHG simulations were retrieved. RefC2 is a transient "future reference" simulation covering the period 1960-2100 with a 10 year spin-up that starts in 1950. The aim of this simulation is to investigate the future evolution of the atmosphere. From 1960 to 2005 GHGs concentrations are prescribed from observations. After 2005, projections of GHGs from the RCP 6.0 scenario are used (Masui et al., 2011). The RCPs are scenarios used to study future Earth's climate. They are composed of four

pathways representative of the GHG concentrations along the 21st century that lead to a radiative forcing of 2.6, 4.5, 6.0 or 8.5 $W.m^{-2}$ in 2100. While RCP 2.6 suppose strong effort to reduce GHG emissions, RCP 8.5 is based on large GHG emissions, with $CH_4$ concentrations being particularly high in this scenario compared to others.

ODS concentrations are prescribed according to the A1 scenario for halogens (WMO, 2011). senC2rcp26, senC2rcp45 and senC2rcp85 are similar to refC2 but instead of following RCP 6.0 for GHGs, they follow RCP 2.6, 4.5 and 8.5 respectively (Meinshausen et al., 2011). The senC2fODS and senC2fGHG are sensitivity simulations, they are similar to refC2 but with ODSs or GHGs fixed at their respective 1960 levels. The senC2 simulations were optional for the intercomparison exercise. Therefore only a few models provided results for both senC2fGHG and senC2fODS experiments (Table 2). A complete description of all CCMI-1 simulations is given by Eyring et al. (2013) and Morgenstern et al. (2017). A summary of these simulations and scenarios along with references is presented in Table 3.

The horizontal and vertical grids vary between the CCMI models. Therefore, all of the required CCMI data are interpolated to a 2° by 2° grid with 86 pressure levels, the highest pressure level being 0.001 hPa. There were 18 models participating in the CCMI simulations. It was thus not possible to perform the same number of UV radiation projections for the entire 21st century due to computational limitations. We choose to only consider the ensemble model median. The error associated with this simplification on the UV radiation projections is discussed in Section 4.1.

From these CCMI simulations, we used the following monthly global fields to calculate UVI: TOZ, OP, TP, $NO_2$, ALB and either altitude or pressure. Some parameters, needed for UV irradiance modeling, were not present in every CCMI model output or just missing from all CCMI model output; Due to the lack of UV albedo in the CCMI model output, broadband albedo is used instead. For $TNO_2$ we vertically integrated the volume mixing ratio of $NO_2$. As single scattering albedo (SSA) is not available, we choose here to use the latest global aerosol monthly climatology, which include global monthly SSA climatology, from Kinne et al. (2013) as input for the TUV model. We used the median AOD and the Ångström exponent (440-870 nm) from three models that provided these variables: CHASER MIROC-ESM, MRI-ESM1r1 and GEOSCCM. While the mean value may be more representative in general, the median was used to avoid possible local erroneous values. Due to the lack of reliable data, total column sulphur-dioxide ($TSO_2$) was set to zero. Nonetheless $TSO_2$ could be an important factor of UVI variability (Zerefos et al., 1986).

Radiative transfer modelling in cloudy conditions is still a challenging task. Bais et al. (2011) used cloud modification factor along with UV irradiance projections in order to simulate future UV radiation changes due to clouds. Here, our focus is on the UV radiation evolution for distinct RCP scenarios and on the influence of GHGs and ODSs. In addition, clouds and aerosols remain the main sources of uncertainties in climate projections (IPCC, 2013), and the accuracy of UV radiation modelling depends strongly on the accuracy of the input parameters. For these reasons, we choose here to analyse only clear-sky conditions. Furthermore, there is also the uncertainty on the absolute mean value of the extra-terrestrial solar UV spectrum used at the top of the atmosphere in TUV. Differences between proposed solar UV spectra can reach 5% (Meftah et al., 2016).

A few other simplifications were made to reduce computational time. OP and TP are averaged zonally but still vary through the 21st century. The distribution of ozone is mainly zonal, and in particular the altitude of the maximum concentration or the maximum concentration has a mainly zonal distribution. On the other hand, the vertical distribution of ozone has a very small

effect on surface UVI compared to TOZ or AOD. Therefore we conclude that the use of a zonal mean introduces only a minor effect on UV radiation calculations. It is reasonable to neglect this compared to other uncertainties associated with the method. For each CCMI monthly output, we simulated UV irradiance at local solar noon and for the 15th of each month. Despite these simplifications, the present study includes the evolution of multiple parameters affecting UVI for four different RCP scenarios along the 21st century. We will also conduct multiple experiment to distinguish the role of either TOZ, GHGs, ODS or AOD on UVI.

## 2.2   UVI modelling cases.

As stated above, we used four RCP scenarios and two sensitivity simulations, but not all models provided these specific runs (Table 2). To ensure that the resulting TUV simulations would be directly comparable with each other, we defined two experiments from two sets of models. These are summarized in Table 2. The first set is composed of models that provided the refC2, senC2rcp26, senC2rcp45 and senC2rcp85 simulations (see Table 2). From this set of models, we can study the impact on UVI from different RCP scenarios (experiment 1, EXP1). Each model in this set provided simulations that cover 2000-2100 at minimum. The second set is composed of models that provided refC2, senC2fODS and senC2fGHG simulations. This set allows us to investigate the impact of fixing GHGs or ODSs on UV irradiance from 1960 to 2100 (experiment 2, EXP2). We also designed a third experiment (EXP3), based on the models used in EXP1 and designed to study the role of AOD on UVI along the 21st century. In EXP3 we performed three simulations; the first one with transient TOZ and transient AOD (hereafter EXP3A), a second with TOZ fixed at its 2000s decadal mean value and transient AOD (EXP3FTOZ), and the last one with AOD fixed at present-days climatological values (Kinne et al., 2013) and transient TOZ (EXP3FAOD). For each experiments we calculated the mean and median of the various input parameters for the selected models, such as ozone, temperature or ground albedo, and used it as input for the radiative transfer model to obtain $UVI_{MEAN}$ and $UVI_{MEDIAN}$.

## 3   Model Validation

In this section, we first investigate the usage of CCMI model data as input for the TUV radiative transfer model. The results are compared against present-day climatological values of UV irradiance obtained from ground-based and satellite measurements. According to Koepke et al. (1998), the UVI modelling error from TUV is about 5 % for a coverage factor of 2 standard deviation.

We gathered UVI data spanning at least the period from 2000 to 2017 for six stations representing six latitudinal bands. The various stations and their characteristics are presented in Table 4. They are all part of the Network for the Detection of Atmospheric Composition Change (NDACC) (De Mazière et al., 2018). UV irradiance measurements at these stations are made by spectroradiometers. Just like UVI obtained by the model, UVI is obtained from the spectral irradiance measurements. These types of measurements have an uncertainty of about 5%. All of these stations began measuring UV irradiance in the early 2000s, except for Reunion Island where observations started in 2009. In order to compare the ground-based measurements to our modelling results, we filter cloudy conditions with the clear-sky flag provided with each station's measurements. We also

select data with a SZA as close as possible to the SZA at local noon, with no more than 2.5° difference. A 10 day average around the 15th of each month was made in order to be consistent with satellite data and avoid numerous missing values. From this we derive a monthly climatology for the 2005-2017 period (UVI$_{GB}$). From the closest grid point of the UVI$_{MEAN}$ and UVI$_{MEDIAN}$ simulation, we derive the same UVI monthly climatology. We do this only for the refC2 simulation. TUV

calculations are made at sea level, stations measurements are made at an altitude ranging between 8m for Barrows up to 370m for Lauder. Mauna Loa is an exception with measurements made at 3397m above sea level (asl).

We also derive a climatology for each station from the OMI OMUVBd product (Krotkov et al., 2002), which is represented by the orange curve in Figure 1 and it will be called hereafter UVI$_{OMI}$. OMUVBd is a level-3 daily global gridded UV-B irradiance product derived from the Ozone Monitoring Instrument (OMI), which is a nadir-viewing spectrometer. Measurements started

in 2004. The instrument covers the spectral region 264-504 nm. The algorithm used to compute surface spectral UV irradiance is the TOMS Surface UV-B flux algorithm (Tanskanen et al., 2007). OMUVBd has previously been evaluated against ground based stations. Tanskanen et al. (2007) found a median overestimation of 0 to 10% of the erythemal doses calculated by OMI against ground-based measurements. Jégou et al. (2011) found a 12.8 ± 3.6 % mean relative difference between OMI clear-sky UVI measurements and ground-based measurements made at the SIRTA observatory (Palaiseau, France) in 2008

and 2009. Brogniez et al. (2016) also analysed this product against three ground-based stations located at Villeneuve d'Ascq and the Observatoire de Haute-Provence, both in metropolitan France, and at Saint-Denis in Reunion island. They observed a systematic overestimation of UVI in the range of 4 to 8% at solar noon. Once more we select UVI only at local solar noon, which is provided in the OMUVBbd product. In order to be as close as possible to our simulation and since OMUVBDd has sometimes missing values over the ground based stations at the 15th of each month, we also always selected 10 days per

20 month centered around the 15th of each month for ground based and satellite data. The results are presented in Figure 1. We also calculate the mean absolute and relative difference between these monthly climatological UVI and UVI$_{GB}$ ground-based observations. Table 4 summarizes these statistics.

### 3.1 Comparison against ground-based measurements

Figure 1 shows that for every station, the average modelled UVI; UVI$_{MEAN}$ (red curve) and UVI$_{MEDIAN}$ (green curve), are close
to the observed climatological UVI (UVI$_{GB}$ black curve). UVI calculated from the single models (light blue curves) are spread around the observations. UVI$_{OMI}$ tends to be a slightly higher than the observations. As found in previous studies, absolute and relative difference (Table 4) shows that UVI$_{MEAN}$ and UVI$_{MEDIAN}$ are always closer to ground-based observations than UVI$_{OMI}$, except for the Palmer station.

At mid and high latitudes, the differences between modelled UVI (UVI$_{MEAN}$ and UVI$_{MEDIAN}$) and UVI$_{GB}$ are consistent. For
UVI$_{MEAN}$, they range between 2.5% and up to 15.8% at Villeneuve d'Asqc and Lauder station respectively. For UVI$_{MEDIAN}$, they range between 2.0% and up to 13.5% at Barrow and Lauder station respectively. While the relative difference can be large at high latitude stations, the absolute difference in UVI is small. For instance, the 10.6 % relative difference in UVI$_{MEAN}$ at the Palmer station, translates into an absolute difference of about 0.3 UVI units. We have to be careful when we interpret UVI at high latitude stations, as the magnitude of UVI is quite small most of the time due to large solar zenith angles. Palmer station

presents the highest differences between $UVI_{MEAN}$, or $UVI_{MEDIAN}$, and $UVI_{GB}$ of about 10%. As presented in Figure 1, there is a large spread between individual models during the beginning of summer (September to December). In addition, we do not observe the same differences at the same season and at the same latitude in the Northern Hemisphere (Barrow). The large spread at Palmer station could be linked to the temporal and geographical representation of the ozone hole.

In the southern tropics, at Saint-Denis, both $UVI_{MEAN}$ and $UVI_{MEDIAN}$ slightly underestimate $UVI_{GB}$ by 3.5 % and 5.3 % respectively. These differences are within the UVI modelling uncertainty. In the northern tropics, at Mauna Loa, a relative difference of -12.3% for $UVI_{MEAN}$ is equivalent to a similar absolute difference of UVI (-1.4 UVI units). Mauna Loa station is located at 3397m asl and the UVI modelled was done at sea level. Therefore, it is coherent to find both $UVI_{MEAN}$ and $UVI_{MEDIAN}$ underestimating the measured UVI. Between $UVI_{MEAN}$, $UVI_{MEDIAN}$ and $UVI_{GB}$, standard deviation of the absolute

difference are the highest for Mauna Loa, at about 0.8 UVI units. The large standard deviation (Table 4) and differences (Figure 1) observed at Mauna Loa are due to the strong difference between modelled and measured UVI during summer. Both modelled UVI and $UVI_{OMI}$ are not able to reproduce ground-based measurements of UVI for this period. For the other stations, the standard deviation ranges between 0.2 and 0.4 UVI units.

      Another factor could explained the differences observed at Mauna Loa and Palmer station. For short time scales of about 10

15   years, a part of the TOZ variability observed at ground-based stations is due to the stratospheric circulation above the station (Poulain et al., 2016). In the refC2 runs, models produce their own wind and temperature fields. In a separate simulation, which we do not analyse here, the refC1SD simulation, the model is nudged towards meteorological reanalyses (Eyring et al., 2013). Unlike refC1SD, refC2 simulations are not designed to reproduce the interannual variability and trends in stratospheric dynamics (and hence ozone) that are observed over individual stations between 2000-2017. Differences between observed

and simulated dynamical variability is thus possibly a significant source of the discrepancies between observed and modelled UVI, that is difficult to verify. The differences in the dynamics of the models (Eichinger et al., 2018) certainly contribute to the spread in the model results. Altough by using simulations with specified dynamics, such as refC1SD, a better agreement may be expected for the validation of CCMI models, the main objective of this study is to study the UVI evolution during the 21st century, which is not possible using refC1SD simulation; therefore we choose to only validate the result from the refC2

simulations.

## 3.2   Comparison against global satellite measurements

$UVI_{OMI}$ tends to overestimate $UVI_{GB}$ by a range that goes from 6.8% at Lauder, and up to 29.3% at Barrow. Standard deviation ranges between 5.0 at Saint-Denis and up to 10.9 at Lauder. $UVI_{OMI}$, which is computed at sea level, underestimate $UVI_{GB}$ at Mauna Loa, which is located at 3306m.

Modelled UVI has also been compared globally to $UVI_{OMI}$ satellite measurements. The relative differences between each model and $UVI_{OMI}$ are represented in Figure 2. To compute the relative differences, we considered the differences between months and then averaged over the entire period and over the globe. While we recognize that this is not the best approach, due to latitudinal and seasonal differences, it will indicate the global differences in the behaviour between the different models. It is not intended to compare the results against $UVI_{OMI}$, but rather to estimate the overall behaviour of the models towards

OMI measurements in order to infer their homogeneity. Over the globe, $UVI_{MEAN}$ and $UVI_{MEDIAN}$ deviate from OMUVBd observations by respectively -16.8 $\pm$ 12.9 % and -17.3 $\pm$ 12.5 %, showing that the response is quite different amongst the individual models. While the closest mean relative difference is observed for the MOCAGE model, that is also the model with the highest variability. In all cases, UVI computed from CCMI models are lower than $UVI_{OMI}$. As stated before previous studies on $UVI_{OMI}$ validation against ground-based spectral measurements found a systematic overestimation. Therefore, in the present study, it is coherent to find lower values of simulated UVI compared to $UVI_{OMI}$.

As a last test, we took the TOZ fields from the 18 models that performed a refC2 simulation from 2000 to 2010 and used them as input for TUV. From there we obtained 18 UVI fields covering the same period and calculated the median, hereafter $UVI_{ALLM}$. The average (over the period 2000 to 2010) of the relative difference between these two data sets ($UVI_{ALLM}$ and $UVI_{MEDIAN}$) is presented in Fig. 3. This result allows us to assess the sensitivity of the radiative transfer model to different ozone inputs. Due to seasonal and inter-model differences, this approach has limitations but the goal of this test was simply to have a first idea of the global difference of behaviour between models. Between both UVI fields there is a mean relative difference of 0.2 $\pm$ 1.9 %. Around the globe, the differences range from -2% up to 2%. In the end, we can say that averaging the CCMI TOZ fields prior to using them as input for TUV induce only a small difference in the resulting average UVI.

$UVI_{MEDIAN}$ and $UVI_{MEAN}$ compare well to the ground-based observations (Figure 1) and have the lowest dispersion among the different models (Figure 2). We therefore concluded that it was reasonable to calculate the UVI from the different simulations based on the median input fields derived from the available models. For this reason, and due to computational constraints stated previously, in the next section we will only discuss the $UVI_{MEDIAN}$ field.

## 4  UVI Projection throughout the 21st century

In the following subsection (4.1), we will discuss the evolution of UVI and TOZ over the 21st century for six latitudinal bands and for the four RCP scenarios by analysing the results of EXP1. We will then (section 4.2) look at the zonal monthly difference in UVI and TOZ between the 2000s and 2090s. In Section 4.3 we evaluate the impact of GHGs and ODSs on the evolution of UVI and TOZ in EXP2. Again, we will start by looking at the percent change of UVI and TOZ from 1960 to 2100. We then investigate the differences between the 1960s and 2090s.

### 4.1  Temporal evolution of UVI during the 21st century according to different RCPs

To investigate the evolution of UVI and TOZ throughout the 21st century, we choose the following latitudinal bands. Southern and northern high latitudes are defined from 90° to 60° S and 60° to 90° N, respectively. Southern and northern mid-latitudes are defined from 60° to 30° S and 30° to 60° N, respectively. Finally, northern and northern tropical latitudes are defined from 30° to 0° S and 0° to 30° N, respectively. We then calculate the zonal mean percent change in the 2090s compared with the 1960s for the four RCP scenarios. These results are presented in Figure 4. Relative percent changes between the 1960s and 2090s are summarised in Table 5 for all latitudinal bands. In order to compare our results to previous studies we also reported results from Bais et al. (2011) and Hegglin and Shepherd (2009).

Figure 4 shows, as expected, that negative changes in UVI are usually correlated with positive changes in TOZ, and vice versa, except in the northern mid and tropical latitudes at the end of the 21st century (section 4.4).

### 4.1.1 Polar regions

In the southern polar region (Fig. 4f), we observe the well known decrease of TOZ that is due to ODS. The ozone layer starts to recover around 2000. Between 2000 and 2100 there is a 10 % increase of TOZ for RCP 2.6 and a 16% increase for RCP 8.5. Consequently, there is a significant decrease of UVI, between 16 to 26 % for these scenarios between 2000 and 2100. Generally, the higher the radiative forcing, the more pronounced are the TOZ increase and UVI decrease. Compared to the 1960s, UVI will still be higher in 2100 by approximately 6.7%, 5.7% and 3.9% for RCP 2.6, 4.5 and 6.0, respectively. Only RCP 8.5 allows a complete return of UVI values in this region. For this region, most of the UVI variability can in theory be explained by the recovery of the ozone layer, as discussed in Section 4.4.

The same behaviour is observed in the northern high latitudes (Fig. 4e), however, the magnitude is weaker. Compared to 1960s values, UVI will be 5.5%, 1.7% and 0.5% higher for RCP 2.6, 4.5 and 6.0 respectively. For RCP 8.5, there is a strong decrease of UVI (7.9%).

### 4.1.2 Mid-latitudes

Southern mid-latitudes are similar to southern high latitudes but with a weaker magnitude of TOZ and UVI percent changes (Fig. 4d). A maximal increase of TOZ of $\sim 9\%$ along with a maximum decrease of UVI of $\sim 12\%$ (Fig. 4d) is observed between 2000 and 2100. Compared to 1960 values, UVI percent changes in 2100 are within 0 and 3% depending on the RCP scenarios. In 2100 for RCP 2.6, while TOZ is slightly lower than its 1960 values ($\sim 1\%$), UVI is higher by $\sim 3\%$. Again, the maximum changes occur for the strongest radiative forcing increase (RCP 8.5). In this scenario, GHG effects are stronger, and consequently there is more ozone in this region and UVI is weaker compared to 1960 values.

In the northern mid-latitudes, while TOZ does not vary more than 1% between 1960 and 2000, we observe a significant growth between 2000 and 2100, $\sim 8\%$ for RCP 8.5 (Fig. 4c). As expected UVI percent changes appear to be anticorrelated with TOZ percent changes between 2000 and 2050, but after 2050 while TOZ still increases, UVI is almost constant, due to effects of AOD changes (see section 4.4).

### 4.1.3 Tropical latitudes

For the southern tropics, TOZ and UVI are well anti-correlated, changes during the 21st century are very small and are confined within 0-3% for the period 2000 to 2100 (Fig. 4b). We observe a decreasing UVI from 2000 to 2050, then an increasing UVI from 2050 to 2100. Nonetheless, in this region, at the end of the 21st century, UVI will still be about 3% higher compared to the 1960s. In the northern tropical band, where TOZ appears to change by no more than 1% between 2000 and 2100, we observe a 2% to 4% increase in UVI during this period. The largest UVI percent change is observed for the lowest change of radiative forcing (RCP 2.6) (Fig. 4a). RCP 8.5 presents either negative changes or a modest increase of UVI and it is also correlated

with an increase of TOZ. Methane emissions are large in RCP 8.5: Morgenstern et al. (2018) found that TOZ increases with increasing methane in CCMI models.

### 4.1.4 Summary and discussion

A similar study was carried out by Bais et al. (2011) within the CCMVal-2 activity. They used the refB2 experiment that used the SRES A1B scenario for GHGs (a scenario close to RCP 6.0). Annual-mean surface UVI percent changes were computed against the 1975-1984 mean. Here we computed changes between 1960-1970 and 2090-2100 (Table 5). We can roughly estimate the UVI percent changes between 1960 and 1980 by looking at Figure 2 of Bais et al. (2011).

Between 1960 and 2100 Bais et al. (2011) observe a ~8% and 1% UVI percent change decrease in the northern and southern high latitudes respectively. Here, for the same period, we only observed a similar decrease (~ 7.9%) in the northern high latitudes for RCP 8.5. For the other scenarios, in this region, we find UVI percent changes between 0.5 to 5.5%. In the southern high latitudes, UVI values are higher than the 1960 baseline for RCP 2.6, 4.5 and 6.0 by 6.7%, 5.7% and 3.9% respectively. For RCP 8.5, there is a complete return of UVI to its 1960 values. In this region, this last scenario also presents the result closest to the one found by Bais et al. (2011).

In the southern mid-latitudes, while Bais et al. (2011) also noted a decrease of UVI (~1%) between 1960 and 2100, we found an UVI increases by 3.4, 2.6, 1.8 % for RCP 2.6, 4.5, 6.0, respectively. For RCP 8.5, we found a 2.3% decrease in UVI. And, while Bais et al. (2011) found a decrease in the Northern Hemisphere for these latitudes, here we show that UVI increases in all scenarios except RCP 8.5. Again, for these two regions, RCP 8.5 shows close result to those by Bais et al. (2011) study.

In the tropical belt (30° North to 30° South), between 2000 and 2100, Bais et al. (2011) found changes in UVI of about 1-2%. We found similar results with smaller values, between 0 to 3%.

Hegglin and Shepherd (2009) conducted a study on UVI changes due to stratospheric circulation-driven changes in the ozone distribution using the Canadian Middle Atmosphere Model (CMAM) simulation performed for the CCMVal-2 intercomparison. By comparing UVI between 1960-1970 and 2090-2100, they observed an evolution at all latitudes close to the one found by Bais et al. (2011). Likewise, they found an increase of UVI in the tropics of about 4%. This was also shown by Butler et al. (2016). To conclude, the CCMVal-2 results from Bais et al. (2011), Hegglin and Shepherd (2009), and our results show similar conclusions for the UVI evolution in the tropics, but our results differ in the northern tropical and mid-latitudes depending on the RCP scenario. As stated before, UVI is influenced mainly by TOZ but also by AOD, ALB, TNO2, OP, and TP. Hegglin and Shepherd (2009) used the analytical formula by Madronich (2007) to determine UVI, which only considers relative changes in TOZ so they could not analyse the effects of other changes. Bais et al. (2011) used a radiative transfer calculations, but aerosol properties were fixed to present climatological values. This could explain the different conclusion obtained in the present study for the Northern Hemisphere, along with the different scenario used (SRES A1B for CCMVal-2, RCPs for CCMI). To better understand the evolution of UVI in the northern mid and tropical latitudes, we will look at the other parameters in section 4.4.

While this section looked at the evolution of UVI throughout the 21st century. In the next section, we will quantify in more detail the difference between the 2000s and the 2090s and between different climate scenarios.

## 4.2 Global UVI levels at the end of the 21st century.

UVI and TOZ zonal monthly differences between the 2000s and 2090s are presented in Fig. 5, for four RCPs for both UVI (left column) and TOZ (right column). There are some missing values during the winter months, because we chose a threshold for the SZA of less than 60 °to calculate the UVI.

First, we note that the strongest mean relative difference (MRD) of UVI or TOZ over the globe is associated with the strongest radiative forcing change. For RCP 8.5, UVI MRD over the globe is -7.9% and TOZ MRD is 6.74%. For RCP 2.6 we calculate a UVI MRD of -1.4 % and a TOZ MRD of 2.1%.

In the northern mid latitudes, while TOZ levels increase with increasing radiative forcing, we do not observe a corresponding decrease of the UVI. Instead a strong increase of UVI is observed for RCP 2.6 for the months of August to November. This increase declines as radiative forcing increasing but keeps its seasonality. This behaviour is not observed in the southern mid-latitudes. We calculate there a decline of UVI associated with a rise in TOZ for all scenarios and for all months.

In the tropics, TOZ decreases for all months and scenarios except for RCP 8.5 where TOZ increases in July.

During the months of September, October and November and in each RCP, there is a strong decrease of UVI (more than 24%) associated with a strong increase of TOZ in the southern polar region from July to November. This is due to the strong recovery of the ozone layer in this region.

The zero line separating a decrease of UVI at high latitudes from an increase at low latitudes appears to shift towards the equator as the radiative forcing increases. Thus, the regions where UVI increases (up to 4 to 6 %) are concentrated around the equator with the increase in radiative forcing that is related to GHG concentrations. This could be explained by the larger GHG concentrations in the RCPs with higher radiative forcing, which are expected to play an important role for the BDC (Butchart, 2014).

In the following sections we will investigate the impact of GHG, ODS and AOD on the UVI separately.

## 4.3 Effects of greenhouse gases and ozone depleting substances on future UVI.

To investigate the effects of GHGs and ODSs on UVI variability between 1960-2100, we analysed the CCMI sensitivity experiments senC2fGHG and senC2fODS. The CCMI models used in this section are those that provided data for the refC2, senC2fGHG and senC2fODS simulations. For the previous experiment (EXP1), we used the median AOD provided by three CCMI models (CHASER-MIROC-ESM, GEOSCCM and MRI) as input for the radiative transfer model. Here, we fixed the AOD by taking the climatological values provided by Kinne et al. (2013). The UVI and TOZ evolution for these two CCMI sensitivity experiments (senC2fGHG and senC2fODS) and refC2 are presented in Fig. 6.

As expected, TOZ shows the smallest trends in the simulations with fixed ODS at all latitudes; the same conclusion can be drawn for UVI. Since the senC2fGHG and refC2 simulations are in close agreement in the Antarctic region, we infer that climate change, linked to GHGs, has the smallest influence on TOZ variation (Dhomse et al., 2018) and therefore on UVI variation in this region.

From these two experiments (refC2 and fGHG), we note that the return of TOZ to 1960 levels will happen later under the fixed GHG scenario, at both northern and southern high latitudes (Fig. 6e,f, orange and red dotted curve).

In the southern mid-latitudes (Fig. 6d), we observe a similar behaviour, with TOZ and UVI percent changes that increase or decrease more rapidly with transient GHG concentrations. This is comparable to what happens in the Northern Hemisphere (Fig. 6c), where GHGs induce a rapid increase of TOZ and a rapid decrease of UVI, which are expected to reach $\sim 3\%$ and $\sim -3\%$ in 2100, respectively.

In the tropics (Fig. 6a,b), ODSs explain about 2% of UVI and TOZ variability. Variations in GHG concentrations appear to have almost no effect on UVI and TOZ until the middle of the 21st century. There is a 2% increase of UVI, which appears around 2070. This can be observed for the fixed GHG and fixed ODS simulations. The percent change in UVI for the refC2 simulation stabilizes around 2070. In this region GHGs are responsible for the acceleration of the BDC, which then induces a decrease of ozone in the lower stratosphere. But they are also responsible for the cooling in the upper stratosphere, which induces an increase of ozone. Therefore, the small magnitude of changes in this region could be explained by the compensating GHGs effects in the simulations (Kirner et al., 2015; Morgenstern et al., 2018).

Global monthly relative differences between the 2090s and 1960s are also plotted in Figure 7 for both, UVI (left column), and TOZ (right column) for the refC2, senC2fODS and senC2fGHG simulations.

With fixed ODS, there is a 3.75% mean relative difference of UVI over the globe, driven by increasing GHGs that affect the circulation. In the tropical belt changes are $\sim 2\%$ higher compared to the standard refC2 run especially in the summer for both hemispheres. Nonetheless, the tropical region is also the place where UVI has the highest absolute values, therefore even a small relative increase means a moderate increase of absolute values. With fixed GHG, the effects of ODSs are minimal when considering the difference between 2100 and 1960.

### 4.4 Other effects affecting UVI.

In Sections 4.1 and 4.2, we discussed UVI increases in the northern mid and tropical latitudes, which were not correlated with TOZ changes. Figure 8 shows the percentage change of UVI, TOZ, and AOD in the northern high, mid, and low latitudes for the EXP3A, EXP3FTOZ, and EXP3FAOD experiments. This is also done for the southern latitudes (Figure 9). In Table 6, we summarize the UVI percent changes between 2100 and 2000 for three EXP3 experiments. We also report the TOZ and AOD changes. All results discussed here refer to zonal averages. Since AOD exhibits large spatial variability, these results should not be generalized to all longitudes.

#### 4.4.1 Polar regions

At northern high latitudes, AOD decreases by $\sim 80\%$ and TOZ increases by $\sim 4\%$ at the end of the century. With transient AOD and TOZ (EXP3A), UVI decreases by $\sim 3\%$ (orange curve, Fig. 8a) and appears to follows the TOZ variability during the 21st century. For this significant decrease of AOD and medium increase of TOZ, UVI are still decreasing. With fixed TOZ (EXP3FAOD), the 80% decrease of AOD result in a 2% increase of UVI. With fixed AOD (EXP3FAOD), the 4% decrease of TOZ result in a 6% decrease of UVI. In this region both TOZ and AOD drive UVI levels.

In the Southern Hemisphere, the situation is different, as shown in Figure 9. AOD changes are very small in this region. The simulations with transient TOZ and either fixed AOD (blue curve) or transient AOD (orange curve) are almost identical and there is almost no UVI change if TOZ is fixed (green curve). UVI changes are driven by the TOZ changes, which are consistent due to the recovery of the ozone layer.

### 4.4.2 Mid-latitudes

At the northern mid-latitudes in EXP3FAOD, UVI decreases and is clearly anticorrelated with TOZ changes (blue curve, Fig.8b). For the same region in EXP3FTOZ, there is a 6% change in UVI in 2100 (green curve, Fig. 8b). In the same figure, in EXP3A, UVI (orange curve) also increases, but by a smaller amount (up to $\sim$ 4% at the end of the 21st century). Both TOZ and AOD drive the UVI variability in this region. As the RCPs project a decline in aerosols precursor emissions (van Vuuren et al., 2011), AOD decreases especially in the Northern Hemisphere and this has a strong effect on UVI.

In the southern mid-latitudes (Fig. 9c,f and i), there are small AOD percent changes and UVI percent changes in the experiment EXP3FAOD (blue curve) and EXP3FA (orange curve) are almost almost identical. We can conclude that TOZ drive the UVI variability in this region.

### 4.4.3 Tropics

In the northern tropics (Fig. 8c,f and i), AOD decreases by $\sim$ 16% and TOZ only changes slightly (<1%). The experiments with transient AOD and either fixed (EXP3FTOZ, green curve) or variable TOZ (EXP3A, orange curve) exhibit a similar percent change in UVI, especially after 2050 when there is a strong decrease of AOD. This would indicate that AOD changes drive the UVI at these latitudes due to the small amplitude of the TOZ variations.

In the southern tropics (Fig. 9c,f and i), where there are small AOD percent changes, the simulations with transient TOZ and either fixed AOD or transient AOD are almost identical. Therefore, TOZ drive the UVI variability in this region.

### 4.4.4 Summary and discussion

In summary, the UVI evolution observed in the Northern Hemisphere (Section 4.1) can be explained by both TOZ and AOD changes (Figure 8). In the Southern Hemisphere, where there are small AOD percent changes, TOZ is the main driver of UVI variability.

Bais et al. (2015) also investigated the impact of AOD and TOZ, along with clouds and surface reflectivity, on UVI. Due to ozone changes, between 2010-2020 and 2085-2095, they found a $\pm$ 2-4% UVI changes in the tropics, a 5-10 % decrease in mid-latitudes and a 40% decrease in Antarctica. We found results close to Bais et al. (2015), with a $\pm$ 1% changes in the tropics, a 3 to 6 % decrease in mid-latitudes and a 25 % decrease in the southern high latitudes. Due to aerosols changes, Bais et al. (2015) also found a strong effect of AOD (10-50%) in the Northern Hemisphere, especially over south east Asia where a $\approx$50% UVI increase is reported. It should be noted that UVI changes due to AOD presents strong longitudinal variability

(Bais et al., 2015) which are not considered in the zonally average results presented here. Therefore the present results can not be generalised.

This last result shows that UVI evolution in the future will not only depends on TOZ but also on AOD. Bais et al. (2015) also addressed this subject and expressed valid concerns on the uncertainties associated with the aerosol effect. In general AOD remains, together with clouds, one of the biggest sources of uncertainty in climate projections (IPCC, 2013). Additionaly, single scattering albedo (SSA), which in this study was fixed at present-day climatological values, has a strong effect on AOD absorption of UVI (Correa et al., 2013). For all theses reasons, to obtain more precise results, future studies should be conducted considering the impacts of clouds, AOD and SSA on future UVI levels.

## 5 Summary

This article focused on the modelling of the UVI evolution throughout the 21st century for multiple scenarios. The objective were to adress the effects of GHGs, ODSs, TOZ and AOD changes on UVI. We have shown that the use of CCMI model data, together with a radiative transfer model (TUV) can reproduce the current climatological values of clear-sky UVI derived from measurements, in most cases, to within a $\pm 5\%$ margin of error. UVI simulated in this way over the globe presents a negative median relative difference compared to satellite observations ranging between 0 to 20%. Compared to ground-based observations, the mean relative difference ranges from -13.2% to 15.8%. In order to compare against the above-mentioned ground-based observations, we have reproduced, using our data, the monthly climatological variability at six stations spread across latitudes.

We also investigated the impact of ODS and GHG on UVI. We have confirmed the role of GHGs in accelerating the return of UVI to 1960 levels via accelerating the ozone recovery. We estimated that GHGs explain approximately 3.8% of the UVI changes between 1960 and 2100. While ODS have an effect on UVI between 1960 to 2050 due to the increased ozone depletion, fixed GHG simulation show small changes of UVI due to ODS changes alone.

In the context of a changing climate, we have considered surface UV irradiance changes in clear-sky conditions and projected globally over the 21st century. However, cloud effects may alter significantly the predicted changes. In this study we investigated the changes for different RCP scenarios (Figure 4).

In almost all scenarios at high southern latitudes, as TOZ returns to 1960 levels, UVI stays 5 to 8% above 1960 levels. Only in RCP 8.5 UVI returns to 1960s values, since it has been shown (Dhomse et al., 2018; WMO, 2014) that TOZ return dates will occur sooner under RCP 8.5. We have found here that UVI levels are mainly driven by TOZ changes at these latitudes, therefore UVI will also return to 1960 levels sooner for RCP 8.5.

At mid-latitudes, UVI changes are not homogeneous. In general, UVI presents higher values in 2100 compared to 1960 in both hemispheres, except with RCP 8.5 where UVI decreases (Table 5). In the Northern Hemisphere, UVI increases are higher during the summer months. In the Southern Hemisphere, between 20°and 30°, there are months without any increase of UVI (Figure 5). The higher emissions of GHGs assumed in RCP 8.5 cause significant differences compared to other scenarios. In the Southern Hemisphere, UVI levels are mainly driven by TOZ, but in the Northern Hemisphere, the declining AOD

(considering the median of three chosen CCMI models) opposes the effect of the TOZ increase. AOD and TOZ are the main drivers of clear-sky UVI variability in this hemisphere with AOD being approximately twice as important as TOZ. Further studies are however needed to investigate this more in-depth. In our present work, only AOD and the Ängström exponent vary during the 21st century, while SSA was fixed to present-day climatological values. Higher values of SSA would increase the absorption effectiveness of AOD and thus impact UV radiation (Correa et al., 2013). Regionally varying SSA changes are expected globally (Takemura, 2012). The upcoming Aerosol and Chemistry Intercomparison Project (AerChemMIP) (Collins et al., 2017) will provide an opportunity to examine this subject.

Zonal mean UVI percent changes from 1960 levels are limited to 0-3% over both tropical bands (0°-30°S and 0°-30°N) during the period 1960 to 2100 (Figure 4). This result is similar to the one found by Bais et al. (2015), Bais et al. (2011) and Hegglin and Shepherd (2009). Increases in the tropical band are higher in the summer of both hemispheres when comparing the decade 2000-2010 against the decade 2090-2100; in this instance local maxima of 8 to 10% were found (Figure 5). An increase of 10% in the tropical UVI would be a matter of concern, since the tropics are already the region with the highest values of UVI, therefore even a small increase could have a strong effect on the biosphere. Regarding human health, considering how important human behaviour can be when assessing human health impact, it might be hard to deduce it only from UVI changes.

The impact of these types of increase on human health, the biosphere and consequently on biogeochemical cycles should be the subject of future studies. Furthermore we recognize that several simplifications were used in the present work, and therefore further analysis on seasonal and longitudinal variabilities along with differences between models should be conducted in order to more accurately assess the impact and uncertainties due to AOD, zonal differences and surface reflectivity on UVI calculations.

| Model | Institution | PIs | References |
|---|---|---|---|
| ACCESS-CCM | U. Melboune, AAD, NIWA | K. Stone, R. Schofield, A. Klelociuk, D.Karoly, O. Morgenstern | Morgenstern et al. (2009), Stone et al. (2016) |
| CCSRNIES MIROC3.2 | NIES, Tsukuba, Japan | H. Akiyoshi, Y. Yamashita | Imai et al. (2013), Akiyoshi et al. (2016) |
| CHASER (MIROC-ESM) | U. Nagoya, JAMSTEC, NIES | K. Sudo, T. Nagashima | Sudo et al. (2002), Sekiya and Sudo (2012), Watanabe et al. (2011) |
| CMAM | CCCma, Canada | D. Plummer, J. Scinocca | Jonsson et al. (2004), Scinocca et al. (2008) |
| CNRM-CM5-3 | CNRM, Toulouse, France | M. Michou, D. Saint-Martin | Michou et al. (2011), Voldoire et al. (2013) |
| EMAC-L90 | DLR, Oberpfaffenhofen, Germany | P. Jöckel, H. Tost, A. Pozzer, M. Kunze, O. Kirner, | Jöckel et al. (2010), Jöckel et al. (2016) |
| GEOSCCM | NASA GSFC, Greenbelt, USA | L. D. Oman, S. E. Strahan | Molod et al. (2015), Oman et al. (2011) |
| HadGEM3-ES | MOHC, UK | F. M. O'Connor, N. Butchart, S. C. Hardiman, S. T. Rumbold | Hardiman et al. (2017), Walters et al. (2014), O'Connor et al. (2014), Madec et al. (2015), Hunke et al. (2010) |
| LMDZrepro | LMD, IPSL, Paris, France | S. Bekki, M. Marchand, F. Lott, D. Cugnet, L. Guez, F. Lefevre, S. Szopa, R.M Hu | Dufresne et al. (2013), Marchand et al. (2012), Szopa et al. (2013) |
| MOCAGE | CNRM, Toulouse, France | B. Josse,V. Marecal | Josse et al. (2004), Guth et al. (2016) |
| MRI-ESM1r1 | MRI JMA, Tsukuba, Japan | M. Deushi, T. Y. Tanaka, K. Yoshida | Yukimoto et al. (2012), Deushi and Shibata (2011) |
| NIWA-UKCA | NIWA, Wellington, NZ | O. Morgenstern, G. Zeng | Morgenstern et al. (2009), Morgenstern et al. (2017) |
| SOCOL | PMOD/WRC, IAC/ETHZ | E. Rozanov, A. Stenke, L. Revell | Revell et al. (2015), Stenke et al. (2013) |
| ULAQ | U. L'Aquila, Italy | G. Pitari, G. Di Genova, D. Visioni | Pitari et al. (2014) |
| UMSLIMCAT | U. Leeds, UK | S. Dhomse, M. P. Chipperfield | Tian and Chipperfield (2005) |
| UMUKCA | U. Cambridge, UK | N. L. Abraham, A. T. Archibald, R. Currie, J. A. Pyle | Morgenstern et al. (2009), Bednarz et al. (2016) |
| WACCM (CESM1) | NCAR | D. Kinisson, R. R. Garcia, A. K. Smith, A. Gettelman, D. Marsh, C. Bardeen, M. Mills | Marsh et al. (2013), Solomon et al. (2015), Garcia et al. (2017) |

Table 1. CCMI Model with Principal Investigator (PIs) and institutions.

| Characteristics | EXP1 | EXP2 | EXP3 | | |
|---|---|---|---|---|---|
| | | | EXP3A | EXP3FTOZ | EXP3FAOD |
| Simulation | refC2 (RCP 6.0) RCP 2.6 RCP 4.5 RCP 8.5 | refC2 (RCP 6.0) senC2fODS senC2fGHG | refC2 (RCP 6.0) | | |
| TOZ AOD | Transient Transient | Transient Fixed (Kinne et al., 2013) | Transient Transient | Fixed (2000-2010 values) Transient | Transient Fixed (Kinne et al., 2013) |
| MultiModelMedian from | CCSRNIES MIROC3.2 CMAM LMDZrepro SOCOL ULAQ | ACCESS-CCM CCSRNIES MIROC 3.2 CHASER (MIROC-ESM) CMAM LMDZrepro NIWA-UKCA UMSLIMCAT WACCM | same as EXP1 | | |

**Table 2.** Characteristics of the experiment conducted in this study:

Simulation: CCMI simulations used in the experiment.

TOZ: TOZ evolution throughout the experiment.

AOD: AOD evolution throughout the experiment.

MultiModelMedian: CCMI Model outputs used for the computation of the median.

| Scenario or Simulation | Details | References |
|---|---|---|
| A1 scenario | A scenario describing a rapid economic growth with a demographic peak in the mid-century. Projections of ODS mixing ratios used in refC2. | WMO (2011) |
| RCP2.6/4.5/6.0/8.5 scenarios | Scenarios that describe possible trajectories of the main factors affecting the climate. 2.6, 4.5, 6.0 and 8.5 stand the radiative forcing expected in 2100. Projections of GHG used in refC2. | Meinshausen et al. (2011) |
| refC2 | CCMI simulation spanning the 1960-2100 period. For ODS, refC2 follows the A1 scenario. For GHG, it uses observations until 2005 then RCP 6.0. | Eyring et al. (2013) Morgenstern et al. (2017) |
| senC2rcp26/45/85 | Same as the refC2 simulation but with GHG changed to either the 2.6, 4.5 or 8.5 RCP scenario. | |
| senC2fGHG | Same as the refC2 simulation but with GHG fixed at their 1960 levels. | Morgenstern et al. (2017) |
| senC2fODS | Same as the refC2 simulation but with ODS fixed at their 1960 levels. | |

**Table 3.** Summary of scenarios and simulations.

| Station | Latitude | Longitude | Relative Difference [%] | | | Absolute Difference | | |
|---|---|---|---|---|---|---|---|---|
| | | | MEAN | MEDIAN | OMI | MEAN | MEDIAN | OMI |
| Mauna Loa | 19.54° N | 155.58° W | -12.3 ± 6.0 | -13.2 ± 6.1 | -3.8 ± 8.5 | -1.4 ± 0.9 | -1.5 ± 0.9 | -0.6 ± 1.1 |
| Saint-Denis | 20.09° S | 55.5° W | -3.5 ± 2.8 | -5.3 ± 2.8 | 13.6 ± 5.0 | -0.3 ± 0.2 | -0.5 ± 0.2 | 1.5 ± 0.7 |
| Villeneuve d'Ascq | 50.61° N | 3.14° E | 2.5 ± 6.3 | 2.8 ± 7.1 | 24.5 ± 6.1 | 0.2 ± 0.2 | 0.01 ± 0.2 | 0.8 ± 0.6 |
| Lauder | 45.04° S | 169.68° E | 15.8 ± 8.9 | 13.5 ± 9.3 | 28.7 ± 10.9 | 0.6 ± 0.3 | 0.5 ± 0.3 | 1.3 ± 0.5 |
| Barrow | 71.32° N | 156.68° W | 2.9 ± 9.5 | 2.0 ± 9.3 | 29.3 ± 5.4 | 0.04 ± 0.2 | 0.02 ±0.2 | 0.6 ± 0.2 |
| Palmer | 64.77° S | 64.05° W | 10.6 ± 3.6 | 10.3 ± 3.2 | 8.1 ± 10.1 | 0.3 ± 0.2 | 0.3 ± 0.2 | 0.2 ± 0.4 |

**Table 4.** Mean UVI relative and absolute difference of the monthly climatology between $UVI_{MEAN}$, $UVI_{MEDIAN}$, $UVI_{OMI}$ to the ground-based measurements ($UVI_{GB}$).

If we define the UVI from ground-based measurements as $UVI_{GB}$, the differences are calculated as:

Relative Difference is defined as: $RD = 100 \frac{UVI_{MEAN} - UVI_{GB}}{UVI_{GB}}$.

Absolute Difference is defined as: $RD = UVI_{MEAN} - UVI_{GB}$.

The same calculation applies to $UVI_{MEDIAN}$ and $UVI_{OMI}$. For Barrow and Palmer station we selected the summer months of June, July and August and December, January and February respectively.

| Region | RCP | | | | RCP 6.0 with | senC2fODS | senC2fGHG | Bais et al. (2011) | | Hegglin and Shepherd |
|---|---|---|---|---|---|---|---|---|---|---|
| | 2.6 | 4.5 | 6.0 | 8.5 | fixed AOD | | | Table 2 | Figure 2 | (2009) |
| 60°- 90 °N | 5.5 | 1.7 | 0.5 | -7.9 | -4.8 | -0.6 | 1.4 | -7.48 | ∼ -8 | -9.1 |
| 30°- 60 °N | 8.3 | 5.2 | 5.0 | -1.4 | -1.9 | 2.3 | 0.7 | -4.10 | ∼ -4 | -3.6 |
| 0°- 30 °N | 2.8 | 2.7 | 2.7 | 0.9 | 2.9 | 6.5 | 0.1 | 0.89 | 1-2 | 3.8 |
| 0°- 30 °S | 2.6 | 2.9 | 2.9 | 1.5 | 3.0 | 6.6 | 0.5 | | | |
| 30°- 60 °S | 3.4 | 2.6 | 1.8 | -2.28 | 0.3 | 3.7 | 1.7 | -4.16 | ∼ -1 | 0. |
| 60°- 90 °S | 6.7 | 5.7 | 3.9 | 0. | -2 | -0.1 | 2.7 | -9.8 | ∼ -2 | 3.2 |

**Table 5.** Percent changes in UVI between 2100 and 1960.

Results for RCP 2.6, 4.5, 6.0 and 8.5 are obtained from EXP1 experiment.

Results for RCP 6.0 with fixed AOD, senC2fODS and senC2fGHG are obtained from EXP2 experiment.

Bais et al. (2011) first column result are obtained from Table 2 of the corresponding study, these are percent changes between 2090-2099 and 1975-1984.

Bais et al. (2011) second column result are obtained by a rough estimate of percent changes between 2100 and 1960 from Figure 2 of the corresponding study.

| Region | UVI | | | TOZ | AOD | Comments |
|--------|-------|----------|----------|------|------|--------------------------|
| | EXP3A | EXP3FTOZ | EXP3FAOD | | | |
| 90 - 60 N | -2.1 | 2.1 | -5.5 | 4.5 | -78 | TOZ and AOD drive UVI levels |
| 30 - 60 N | 3.8 | 6.2 | -2.8 | 3.0 | -77 | TOZ and AOD drive UVI levels |
| 0 - 30 N | 3.5 | 2.3 | 1.2 | -0.5 | -15 | TOZ and AOD drive UVI levels |
| 0 - 30 S | 0.6 | 0.0 | 0.5 | 0.2 | -0.3 | TOZ drives UVI levels |
| 30 - 60 S | -6.1 | 0.1 | -6.0 | 5.6 | -4.16 | TOZ drives UVI levels |
| 60 - 90 S | -26.8 | -3.2 | -26.7 | 11.8 | -1.5 | TOZ drives UVI levels |

**Table 6.** Percent Changes in UVI, TOZ and AOD between 2100 and 2000 for the EXP3A, EXP3FTOZ and EXP3FAOD experiments.

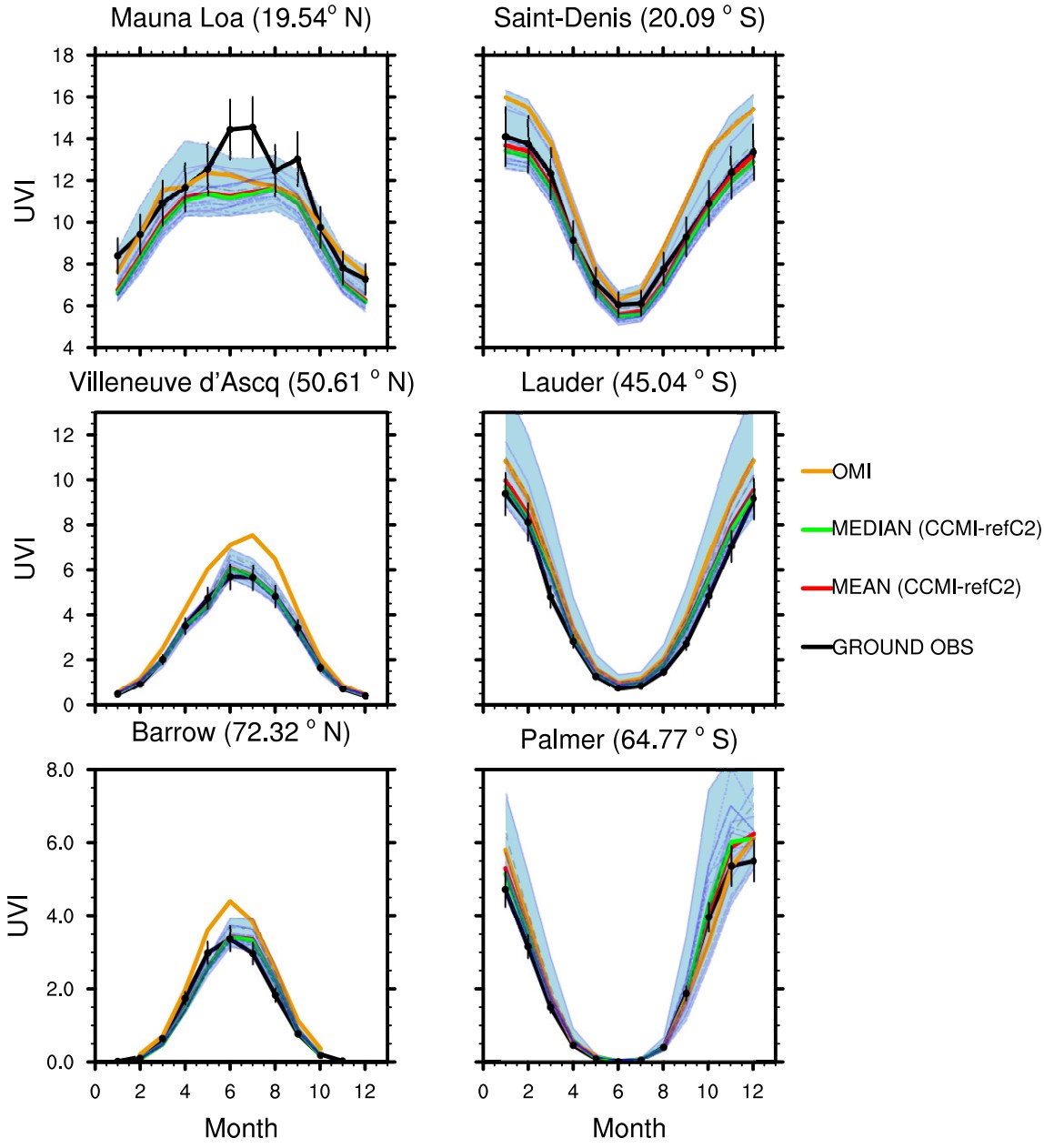

**Figure 1.** UVI$_{GB}$ (2000-2010) for six NDACC stations along with the respective closest grid point from CCMI&TUV UVI simulation (UVI$_{MEAN}$ and UVI$_{MEDIAN}$). Station measurements are represented in the black curve with a $2\sigma$ dispersion bar. UVI$_{MEAN}$ and UVI$_{MEDIAN}$ are represented in green and red. Each CCMI models are represented in light blue, the shaded blue area represents the spread of the models. UVI$_{OMI}$ from the OMUVBd product are in orange.

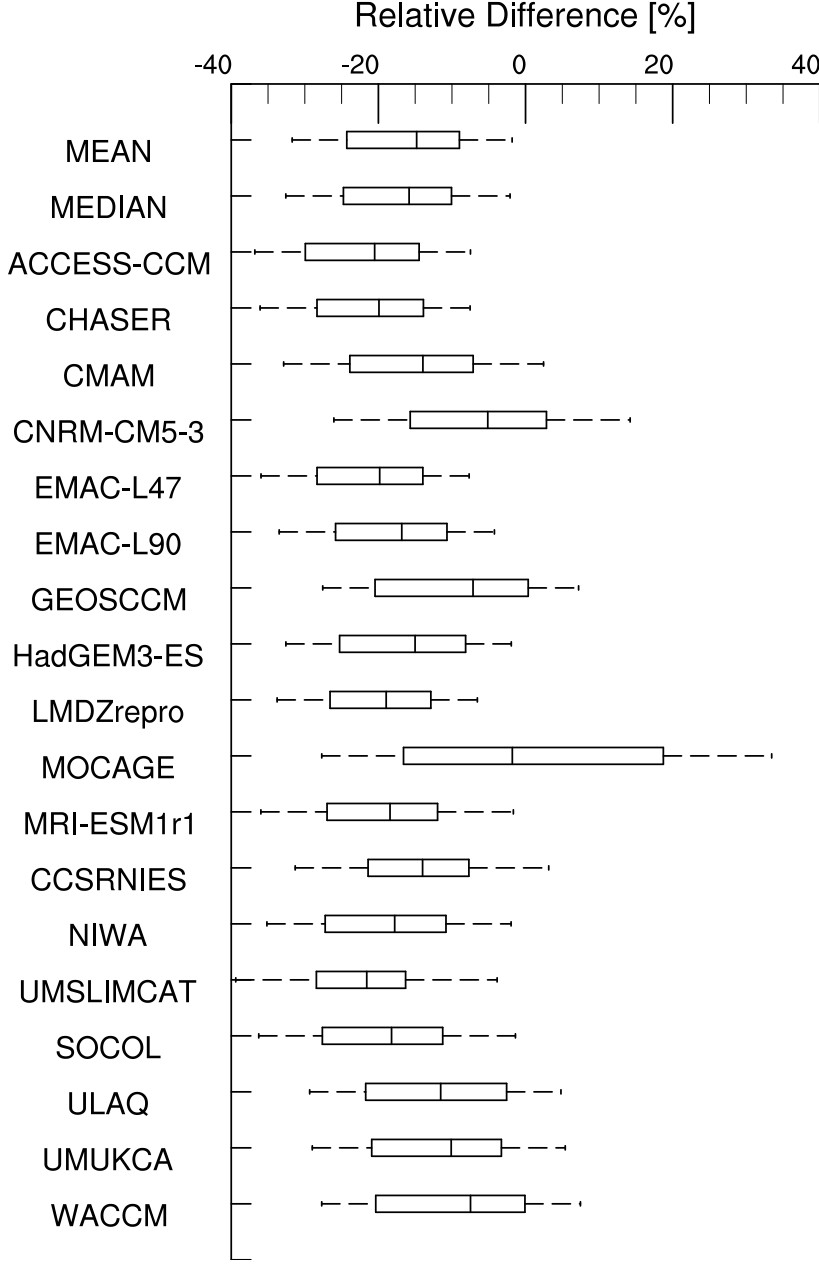

**Figure 2.** Boxplots summaries of the relative differences between the monthly UVI from CCMI models (refC2) and the monthly mean OMUVBd product for the period 2000-2010. Left and right end of the box are the first and third quartile respectively. The line inside the box is the median or second quartile. Left and right end of the whiskers are the mean $\pm$ 1-standard deviation.

For a model M, (M being Mean, Median, ACCESS-CCM, CHASER, ... ) from which we obtained $UVI_M$, we compute:

$$UVI_{RD}[\%] = 100 \times \frac{UVI_M - UVI_{OMI}}{UVI_{OMI}}$$

We then compute the average value of $UVI_{RD}$ over the entire globe and the period 2000-2010.

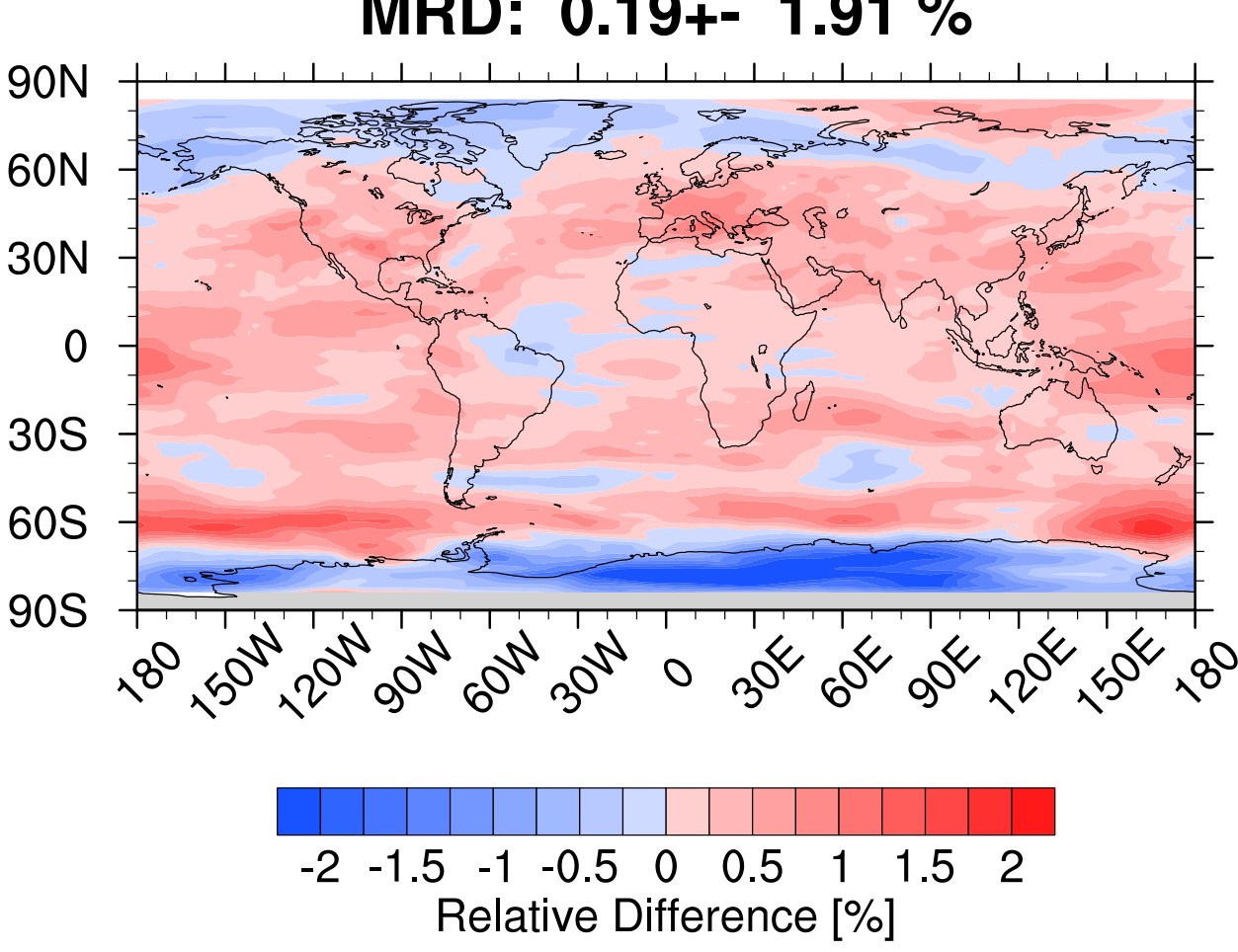

**Figure 3.** UVI annual mean relative difference between the median UVI obtained from the 18 CCMI model data used and TUV and the UVI obtained from the CCMI median TOZ used with TUV for the period 2000-2010.

First, we compute the relative difference:

$$\text{UVI}_{\text{RD}}[\%] = 200 \times \frac{UVI_{\text{ALLM}} - UVI_{\text{MEDIAN}}}{UVI_{\text{ALLM}} + UVI_{\text{MEDIAN}}}$$

Then we compute the average of $\text{UVI}_{\text{RD}}$ over the period 2000-2010 for each point.

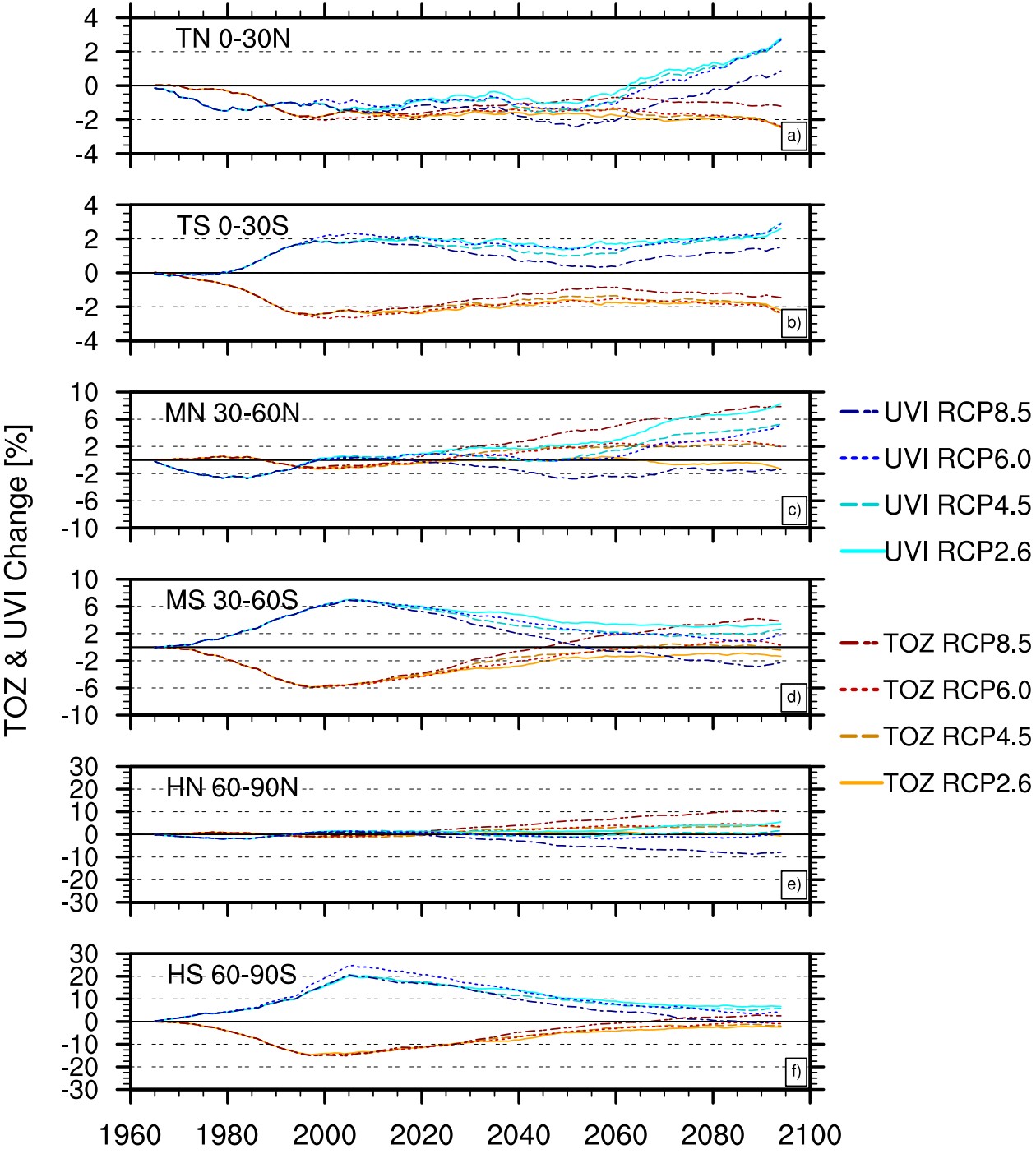

**Figure 4.** UVI and TOZ percent change in 2090-2100 relative to 1960-1970 values for six latitudinal bands; Northern Tropical band (TN 0°-30 °N), Southern Tropical band (SN 0°- 30 °S), Northern Mid Latitude band (MN 30°- 60 °N), Southern Mid Latitude band (MS 30°- 60 °S), Northern High Latitude band (HN 60°- 90 °N), Southern High Latitude band (HS 60°- 90 °S). UVI changes are represented in different shades of blue for the four RCPs scenarios. TOZ changes are represented in different shades of red. Scale of the vertical axis are not the same for each subplot.

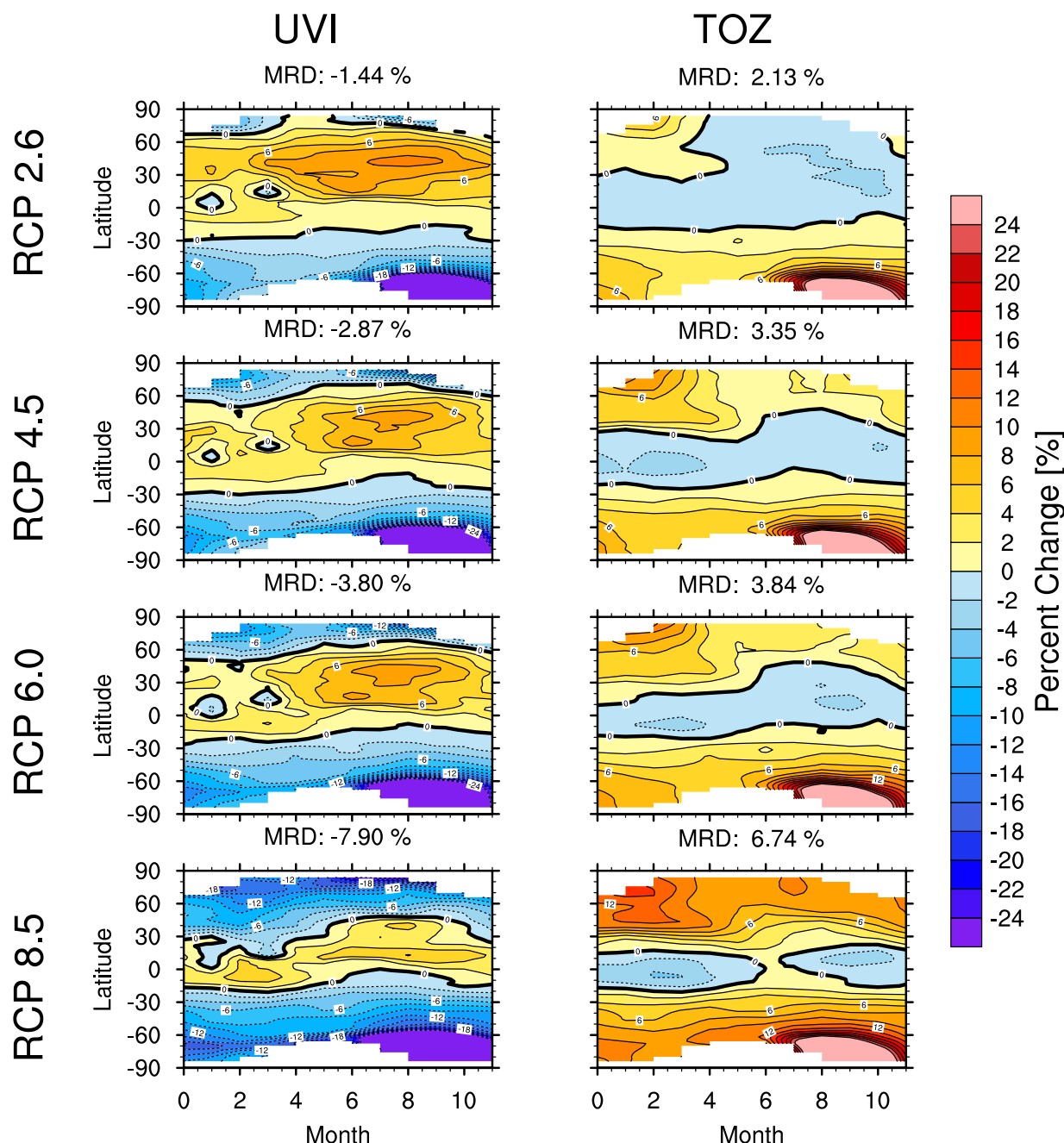

**Figure 5.** Latitudinal and monthly variation of UVI and TOZ percent change in 2090-2100 relative to 2000-2010 for the four RCP scenarios

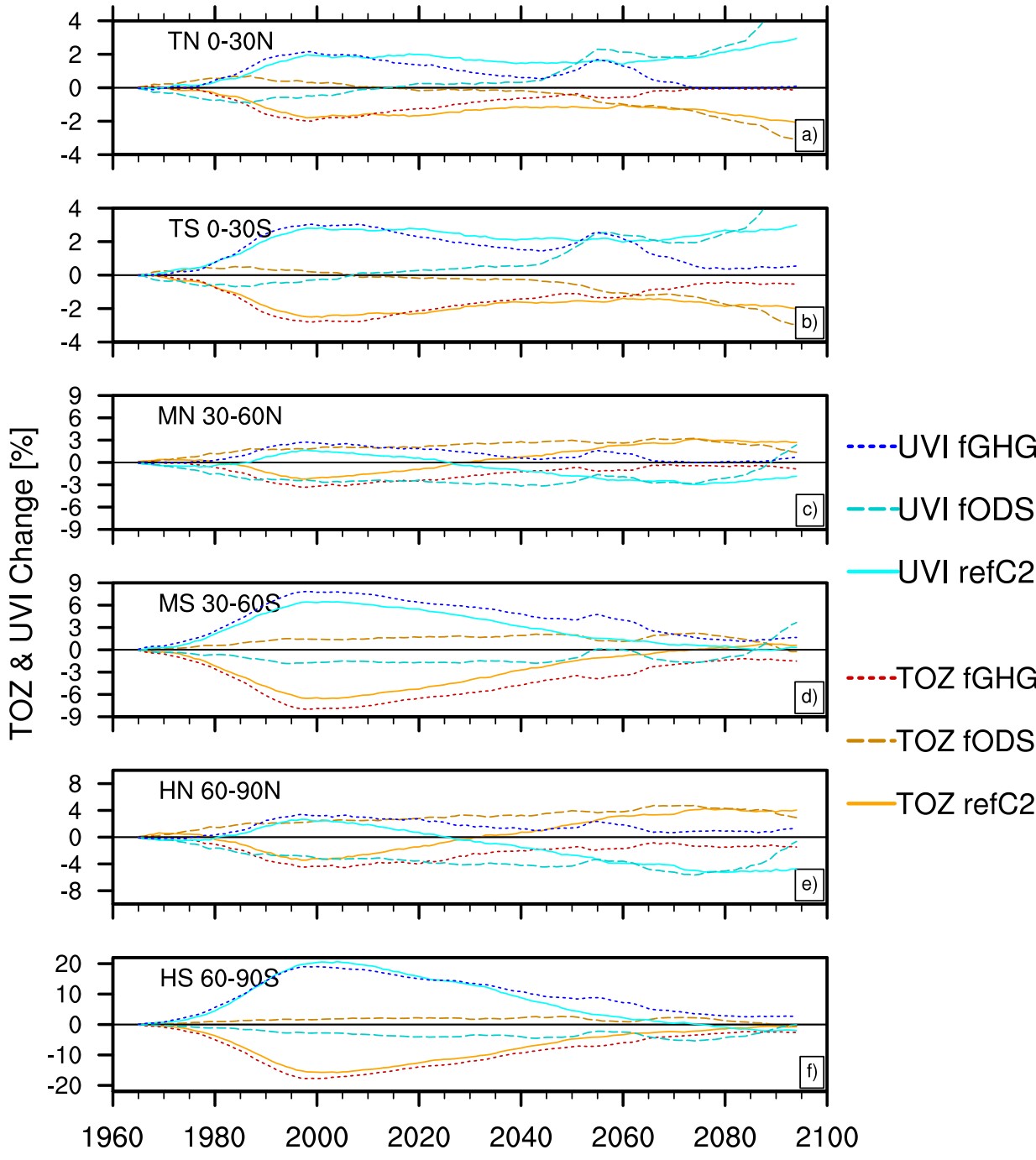

**Figure 6.** UVI and TOZ percent changes (relative to 1960s values) for six latitudinal bands; Northern Tropical band (HN 0°- 30 °N), Southern Tropical band (HS 0°- 30 °S), Northern Mid Latitude band (MN 30°- 60 °N), Southern Mid Latitude band (MS 30°- 60 °S), Northern High Latitude band (HN 60°- 90 °N), Southern High Latitude band (HS 60°- 90 °S). UVI changes are represented in differents shades of blue for the three EXP2 scenarios. TOZ changes are represented in differents shades of red.

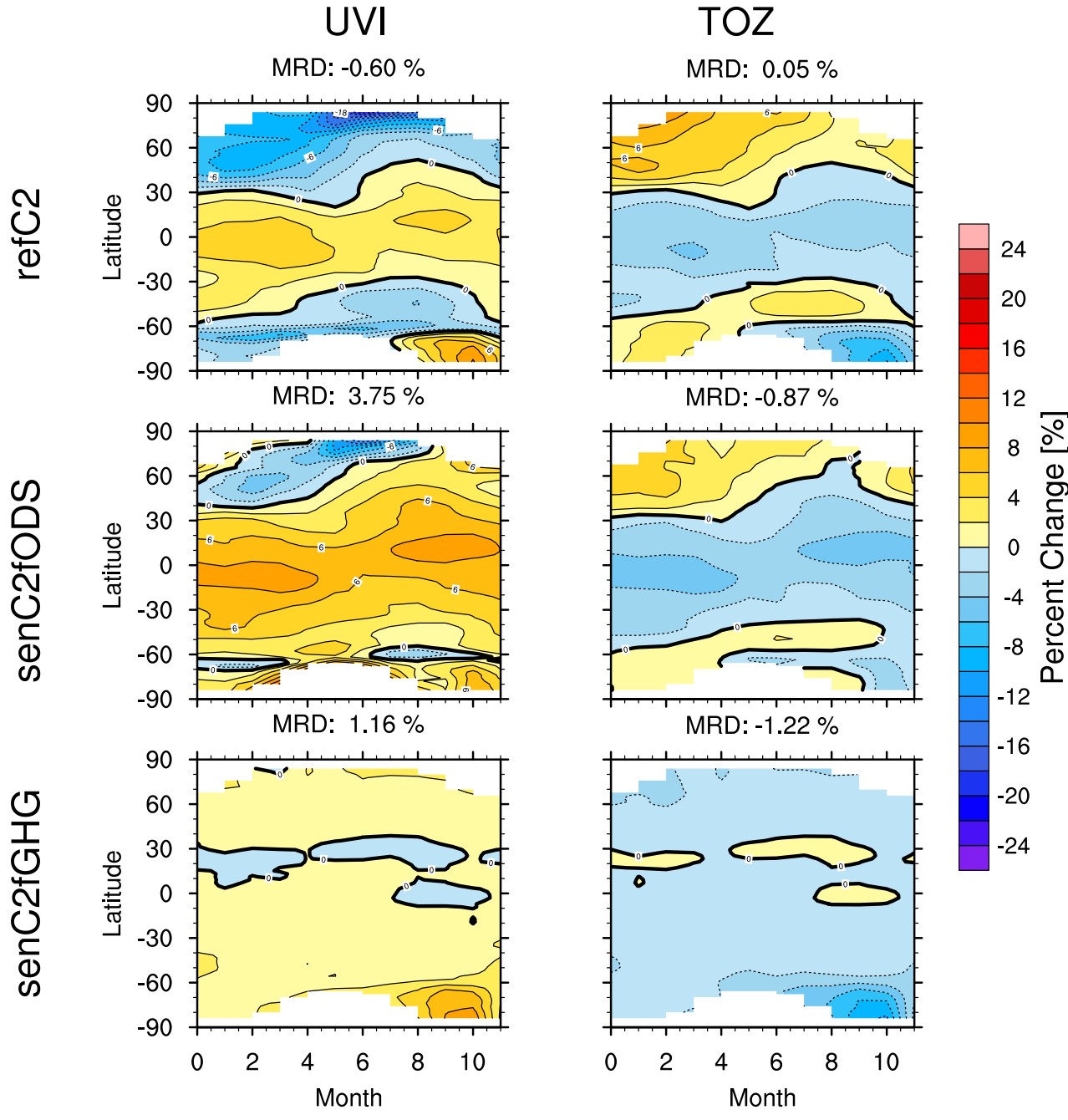

**Figure 7.** Latitudinal and monthly variation of UVI and TOZ percent change between 2090-2100 and 1960-1970 values for refC2, senC2fODS, senC2fGHG.

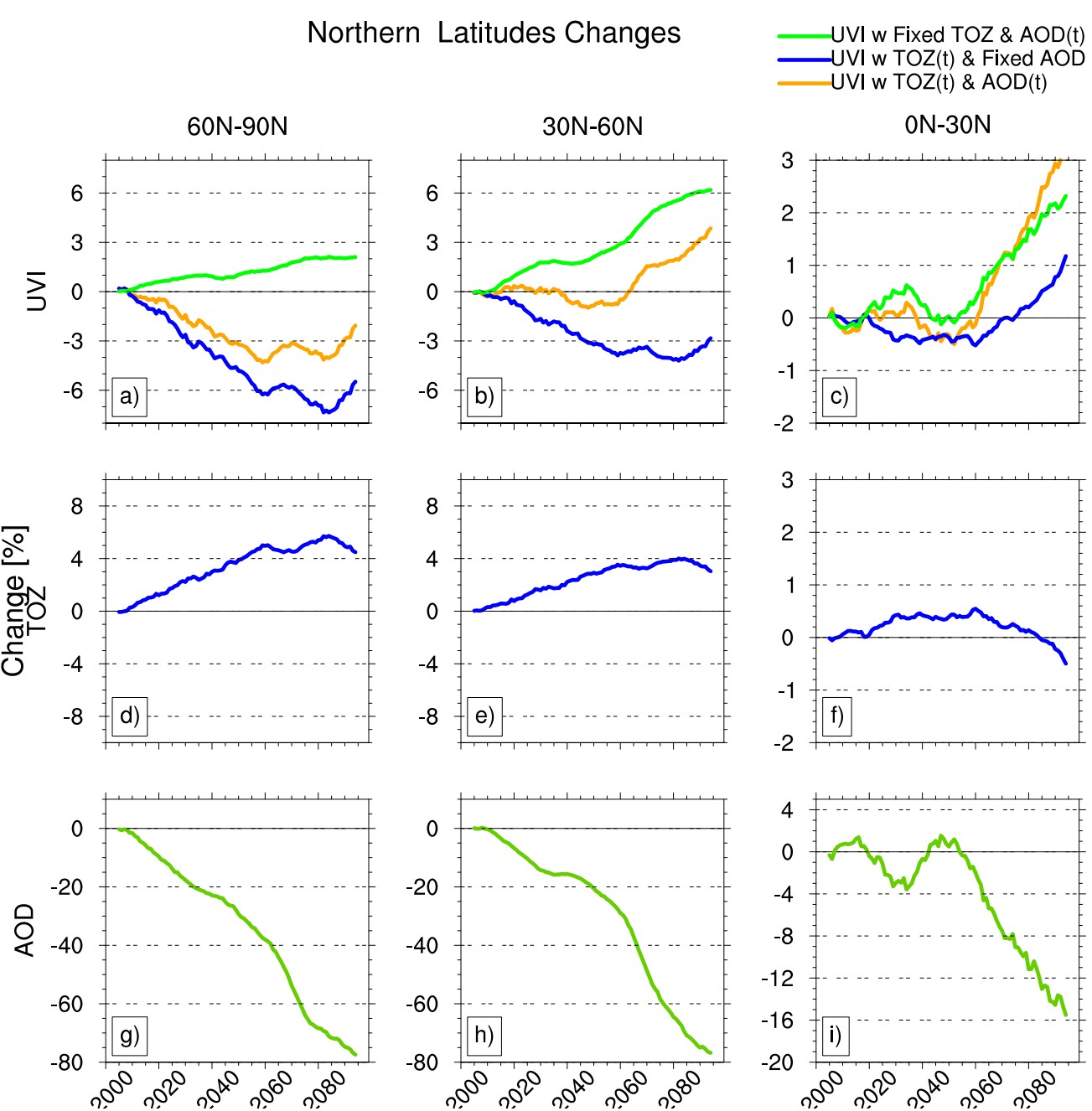

**Figure 8.** UVI, TOZ and AOD percent change from 2000 to 2010 values in the Northern high, mid, and low latitudes for the EXP3 experiment. UVI modelled with transient TOZ and and AOD fixed at present-day climatological values are in blue. UVI modelled with TOZ fixed at present-day climatological values and AOD variable through the 21st century are in green. UVI modelled with transient TOZ and AOD are in orange. TOZ and AOD are respectively in blue and green.

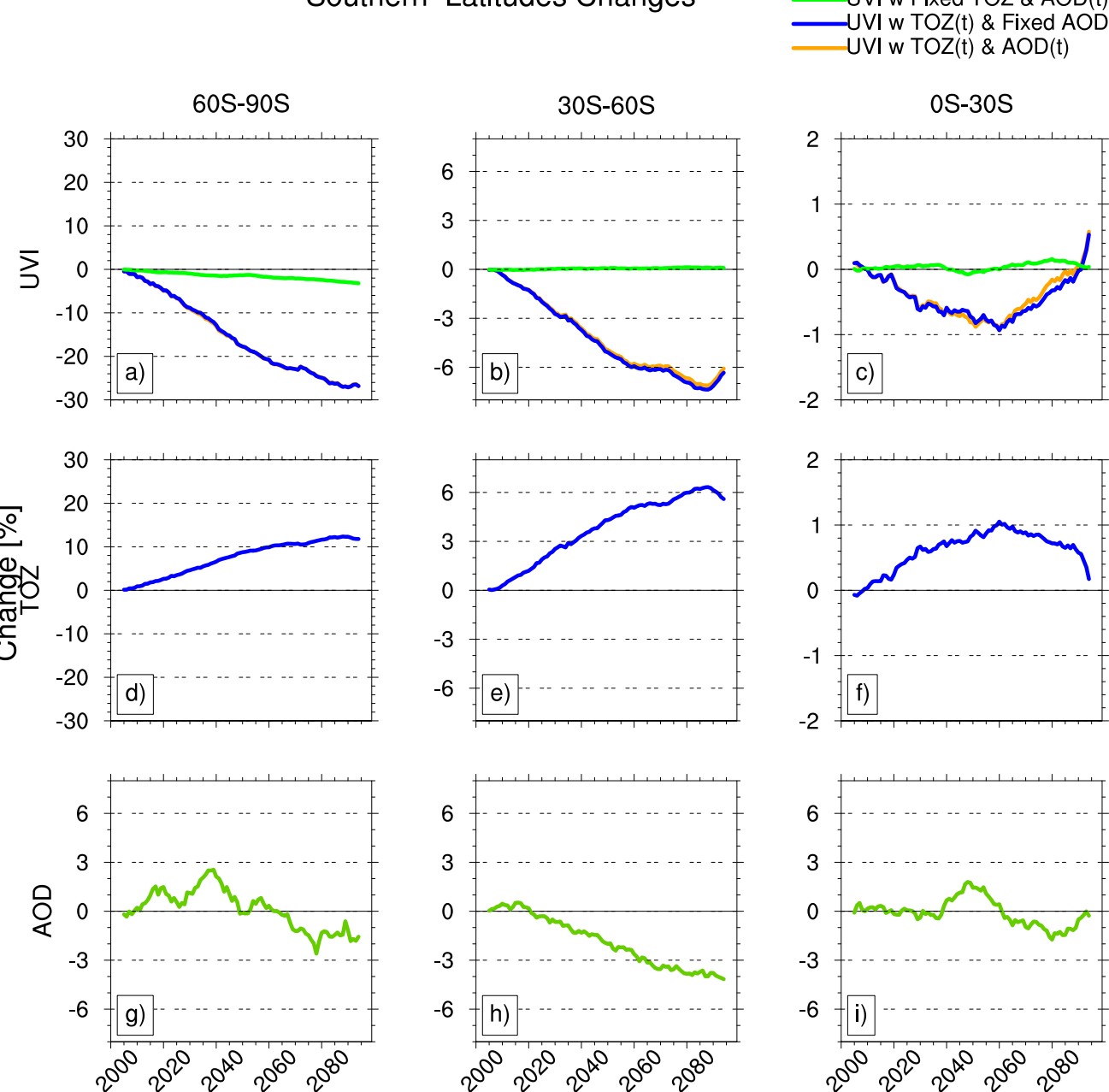

**Figure 9.** UVI, TOZ and AOD percent change from 2000 to 2010 values in the southern high, mid, and low latitudes for the EXP3 experiment. UVI calculated with transient TOZ and and AOD fixed at present-day climatological values are in blue. UVI calculated with TOZ fixed at present-day climatological values and AOD variable through the 21st century are in green. UVI calculated with transient TOZ and AOD are in orange. TOZ and AOD are in blue and green respectively.

*Acknowledgements.* The authors acknowledge the Région Réunion, CNRS and Université de la Réunion for support and contribution within the research infrastructure OPAR (Observatoire de Physique de l'Atmosphère à la Réunion). OPAR is presently funded by CNRS (INSU) and Université de la Réunion and managed by OSU-R (Observatoires des Sciences de l'Univers à la Réunion, UMS 3365).

The data used in this publication were obtained from Brogniez C.(University of Lille 1), Bernhard G. (Biospherical Instruments) and McKenzie, R. (National Institute of Water & Atmospheric Research) as part of the Network for the Detection of Atmospheric Composition Change (NDACC) and are publicly available (see http://www.ndacc.org).

We acknowledge the modelling groups for making their simulations available for this analysis, the joint IGBP-IGAC/WCRP-SPARC
Chemistry-Climate Model Initiative (CCMI) for organizing and coordinating the model data analysis activity, and the British Atmospheric Data Center (BADC) for collecting and archiving the CCMI model output.

This project was supported by the European Project StratoClim (7th Framework Programme, grant agreement 603557) and the grant "SOLSPEC" from the Centre d'Etude Spatiale (CNES).

The SOCOL team acknowledges support from the Swiss National Science Foundation under grant agreement CRSII2_147659 (FUPSOL II). E.R. acknowledges support from the Swiss National Science Foundation under grants 169241 (VEC) and 163206 (SIMA).

CCSRNIES research was supported by the Environment Research and Technology Development Fund (2-1303 and 2-1709) of the Min-
istry of the Environment, Japan, and computations were performed on NEC-SX9/A(ECO) computers at the CGER, NIES.

The EMAC simulations have been performed at the German Climate Computing Centre (DKRZ) through support from the Bundesministerium für Bildung und Forschung (BMBF). DKRZ and its scientific steering committee are gratefully acknowledged for providing the HPC and data archiving resources for the consortial project ESCiMo (Earth System Chemistry integrated Modelling).

N.B., and F.M.O'C. were supported by the Joint UK BEIS/Defra Met Office Hadley Centre Climate Programme (GA01101). N.B. also acknowledges the European Commission's 7th Framework Programme, under grant agreement no. 603557, StratoClim project.

Olaf Morgenstern and Guang Zeng wish to acknowledge the contribution of NeSI high-performance computing facilities to the results
of this research. New Zealand's national facilities are provided by the New Zealand eScience Infrastructure (NeSI) and funded jointly by NeSI's collaborator institutions and through the Ministry of Business, Innovation & Employment's Research Infrastructure programme (https://www.nesi.org.nz). Olaf Morgenstern acknowledges support from the Royal Society Marsden Fund, grant 12-NIW-006, and under the Deep South National Science Challenge.

UMUKCA-UCAM model integrations have been performed using the ARCHER UK National Supercomputing Service and MONSooN system, a collaborative facility supplied under the Joint Weather and Climate Research Programme, which is a strategic partnership between the UK Met Office and the Natural Environment Research Council.

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
