# Peer review of "Clear-Sky Ultraviolet radiation modelling using output from the Chemistry Climate Model Initiative"

_Atmospheric Chemistry and Physics, 2018_

## Referee Comment (RC1) · Anonymous Referee #1 · 2 Jul 2018

General comments: The paper presents the results from modelling of UVI at Earth's surface using the TUV model and data from the first phase of the Chemistry-Climate Model Initiative. The results are for clear-sky only, which is a major drawback in terms of their value for UV predictions to input into understanding risks and benefits for human health and the environment. The findings differ to those of previous studies (Hegglin and Shephard 2009 and Bais 2011), including in the direction of change of UVI at high latitudes. An important finding is that GHGs accelerate ozone recovery and return of UVI to pre-ODS levels, and that the most important driver of UVI in the southern hemisphere is TOZ, but in the northern hemisphere aerosol optical density is around twice as important as TOZ.

There numerous errors throughout the paper in the use of plural vs. singular, e.g.

[Figure]

"The spectral solar irradiance...range from 280-450nm" (should be 'ranges' or revise the sentence and use 'in the range from 280-450nm); "The required input for the UV calculation.." (should be 'inputs'); Page 7, line 12: "these types of measurements has an uncertainty..."(should be 'have'). There are multiple minor errors in English, e.g. page 12 line 20: "which have affect the circulation"

Specific comments: Introduction: 1. General comment: the most recent literature cited is from 2014 – is there no more up to date information that should be used to support the introduction? For example, the recent paper by Dhomse et al (2018) on estimates of ozone return dates is likely to be relevant. 2. Line 9-10. Non-melanoma skin cancer is now being referred to as keratinocyte cancer. Only SCC is caused by chronic exposure; the pattern of exposure for BCC is more complicated, and probably more similar to melanoma, i.e. sunburns particularly during childhood years. The cited reference is very old – 2004. 3. "Studies on human health and UV generally use the UV Index". This statement would be correct only for ecological studies. Individual-level studies would more commonly use some measure of exposure and dose rather than ambient UV irradiance. UVI does not quantify the impact of UV radiation on human skin – this requires a measure of exposure as well as irradiance, and probably for impact, a measure of skin type. The sentence needs to be rephrased – perhaps to note that UVI is the cornerstone of public health messages for sun protection, as a measure of ambient UV irradiance.

Model validation: present day values 1. Page 7-8: this sentence needs to be revised: "UVIOMI tends to be a lightly higher" (slightly?) 2. The comparison between the different UVI measurements is a little confusing – there are the modelled UVI values, the observed from ground-based spectroradiometers, and computed results based on OMI data. At the bottom of page 7 – UVIMean and UVIMedian are the values modelled from the CCMI using TUV; the 'observed climatological UVI' – is the OMI-based estimates? And then 'observations' refers to the measured ground-level data. It would be clearer to consistently use the abbreviations that are used in Table 3, i.e. UVIOMI, UVIMean,

UVIMedian and UVIGB and use those to refer to the different datasets. Also later – simulated, CCMI models, UVIMean – all refer to the same data. 3. Page 8 "The global relative difference between these two data sets.." – it is not clear without looking at Fig 3 which two datasets are under consideration.

Conclusion 1. The final part of the conclusion focuses on possible impacts. For human health, the main driver of health risks is sun exposure behaviour. It is not just a direct association between UVI and health risk – this needs to be noted. 2. As noted in the text, the results differ from those of previous assessments. Much weight is placed on these predictions, in terms of, for example, future health planning. Yet, when the estimates are so different, it is difficult to have any confidence in the findings from one model over another. The authors might address this consideration.

---

## Referee Comment (RC2) · Anonymous Referee #2 · 3 Jul 2018

This paper addresses projections of UV radiation in the 21st century based on projections from Chemistry Climate Models participating in CCMI. The innovation of this study, comparted to previous studies, is the use a new set of projections for ozone and other UV affecting variables.

One important aspect of the study is the investigation of the effects on UV projections under different RCPs.

Overall is an important contribution to quantifying the effects of future atmospheric changes on UV radiation at the surface. However, there are several aspects that need attention and should be clarified before the paper is accepted for publication. Furthermore, the manuscript needs a through language checking.

[Figure]

General comments:

The influence of ozone, aerosols, surface reflectivity and clouds on UV in the past, present, and future were discussed in the 2015 Assessment of the effects of ozone depletion and climate change panel of UNEP (Bais et al., 2015) but this publication is not discussed.

Bais, A. F., R. L. McKenzie, G. Bernhard, P. J. Aucamp, M. Ilyas, S. Madronich, and K. Tourpali (2015), Ozone depletion and climate change: impacts on UV radiation, Photochem. Photobiol. Sci., 14(1), 19-52.

Results of that study are more recent and more relevant compared to those of Hegglin and Shepherd, 2009 and Bais et al., 2011, and should be discussed against the results of this study.

I have some doubts on using mean and median total ozone from all models to derive the UVI projections through the 21st century. By doing this the inter model variability of the projected ozone is lost, and taking into account that UVI is not linearly related to ozone, the projected mean UVI should not equal the mean of the individual projections of UVI. The authors justify their decision to use the mean ozone on the results of Figure 3. However this figure shows annually averaged differences and does not provide any information on the spread of the seasonal differences as well as on the spread of the UVI projections by individual models. These two will determine the uncertainty of the projections and it would be essential to know how large thee uncertainties are.

Unfortunately, if these assumptions are proven incorrect (or if they result into large uncertainties), then the whole discussion on the future evolution of the UVI projections will be questionable.

Specific comments:

P3, L21: Also in Bais et al. 2015

P4, L1: The 11-year solar cycle affects UV (especially UV-B) through changes in stratospheric ozone, while the direct influence is almost negligible.

P5, L6: It would be very informative to show in Table 1, or in a separate table, which of the input parameters are provided by each model and which models have participated in the different experiments used in this study

P6, L4: For which spectral region was the surface albedo provided by the models? Usually the models provide the broadband albedo which is much different than the UV albedo. Please explain how this was handled.

P6, L7: I don't think that taking the median of three numbers is representative for the most likely value of the parameter. Probably in this case the mean optical depth would be more representative.

P6, L16-17: The two references refer to absorption cross sections of ozone and not to solar spectra. Assuming that this is a typo, using in the RTM calculations absorption cross sections different than those used for the retrieval of total ozone (Paur and Bass 1985 for GB instruments) could introduce inconsistency in the results.

P6, L21: Please state which is this simplification, no matter if it is discussed in another section.

P6, L22: Could you provide an estimate of how large would be the effect on the UV calculations due to using zonally averaged profiles?

P6, L27-31: Which models are used in the experiments? This is related to my previous comment (P6, L6).

P7, L7: In the comparison with GB data, has the altitude of each station been taken into consideration in the TUV calculations? UVI at Mauna Loa is by far greater compared to UVI at the seas surface for the same latitude.

P7, L15: In the monthly climatology, were data of the 15th used only (as with satellite data) or the mean of the entire month? How missing data were handled?

P7, L29-31: It is not clear which the satellite data are used. Is always the 10 day average around the 15th is used, or only when data on the 15th are missing? How the clear-sky satellite data are selected?

P7, L31: As it appears from the text, for the monthly GB climatology all available data were used, but for the satellite day only the measurement on the 15th (or the 10-day average). If this is true, the data used in Figure 1 are inconsistent and I do not understand how one can compare these datasets with the model results.

P7, L32- P8, L10: Please check this section and make the discussion clearer. I suggest focusing the discussion on differences between model and GB data and not on differences between GB and OMI because this is not the main subject of this paper. The changes reported in the abstract and the conclusions (-4 to 11%) are not discussed at all in this section. Furthermore it would be good to report in Table 3 the spread of the model differences to GB for each station (e.g. the standard deviation).

P8, L22: Averaging the UV index over the entire globe is not a good approach due to large latitudinal (and seasonal) differences.

P8, L31-34: The results presented in Figure 3 are yearly averaged. However, the variability on predicted ozone varies seasonally (see Dhomse et al., 2018) and differences in UVI may also have a seasonal effect, which now is suppressed. Moreover, the test is performed for a period (2000-2010) when inter-model ozone variations are smaller than later in the 21st century. As changes in the UVI are nonlinearly related to ozone, the assumption that total ozone evolution in the 21st century can be represented by the mean or median is not a safe choice. I suggest checking whether the seasonal behavior of the differences is still within the reported limits, particularly in the southern high latitudes.

P9, L4: Moreover, Figure 3 shows the sensitivity due to averaging of only the total ozone. What is the uncertainty introduced by the averaging of the other input parameters; the ozone and temperature profile, the surface reflectivity, the optical depth of

aerosols?

P10, L17-28: The changes derived in (Bais et al., 2011) have been based on a different reference period (as noted in the text); therefore the results of this study are not directly comparable, particularly as in these early years ozone variations were quite large.

P12, L31-25: Effects of AOD on UVI have large longitudinal variability (see Bais et al., 2105) which is suppressed when taking zonal averages. It would be interesting to compare the effects of TOZ and AOD on UVI of this study with those reported in Bais et al., 2015.

P13, L4: As mentioned above AOD exhibits large spatial variability, therefore the results found for zonally averaged UVI changes should not be generalized for all latitudes. For example, AOD over China will decrease substantially by 2100 but at similar latitudes over the Pacific the effects are almost negligible.

P13, L24: Please discuss to which level of accuracy the UVI can be reproduced.

P14, L1: State here that UVI projections are for cloud-free skies. Cloud effects can alter significantly the predicted changes at high latitudes.

P14, L6: State the period for the increase.

Technical comments:

P4, L14: Replace "increment" with "increase"

P6, L26-27: Replace to: "... with each other, we defined two experiments from two sets of models. These are summarized ..."

P8, L11: I suggest replacing "wind variability" with "stratospheric circulation"

---

## Author Comment (AC1) · 17 Sep 2018

**Response to the Referee's Comments**

September 17, 2018

We would like to thank the referee for his thorough review. The comments have been very beneficial. We hope that the new version has improved the paper (Supplement of this comment).

**1   General comment**

*The paper presents the results from modelling of UVI at Earth's surface using the TUV model and data from the first phase of the Chemistry-Climate Model Initiative. The results are for clear-sky only, which is a major drawback in terms of their value for UV predictions to input into understanding risks and benefits for human health and the environment. The findings differ to those of previous studies (Hegglin and Shephard 2009 and Bais 2011), including in the direction of change of UVI at high latitudes. An important finding is that GHGs accelerate ozone recovery and return of UVI to pre-ODS levels, and that the most important driver of UVI in the southern hemisphere is TOZ, but in the northern hemisphere aerosol optical density is around twice as important as TOZ. There numerous errors throughout the*

*paper in the use of plural vs. singular, e.g. "The spectral solar irradiance.
. .range from 280-450nm" (should be 'ranges' or revise the sentence and
use 'in the range from 280-450nm); "The required input for the UV calcula-
tion.." (should be 'inputs'); Page 7, line 12: "these types of measurements
has an uncertainty. . ."(should be 'have'). There are multiple minor errors in
English, e.g. page 12 line 20: "which have affect the circulation"*

Regarding the general comment and following up on the reviewer's suggestions, we
have looked closely at the level of English of the manuscript, and the cited errors have
been corrected.

**2  Specific comments**

2.1   Introduction

*1.    General comment: the most recent literature cited is from 2014 –
is there no more up to date information that should be used to support the
introduction?  For example, the recent paper by Dhomse et al (2018) on
estimates of ozone return dates is likely to be relevant.*

Bais et al. (2015) and Dhomse et al. (2018) have been cited to support the introduction.

*2.    Line 9-10.  Non-melanoma skin cancer is now being referred to as
keratinocyte cancer. Only SCC is caused by chronic exposure; the pattern
of exposure for BCC is more complicated, and probably more similar to
melanoma, i.e.  sunburns particularly during childhood years.  The cited
reference is very old – 2004.*

We thank the referee for his input, more details on this subject and recent references have been added.

3. *"Studies on human health and UV generally use the UV Index". This statement would be correct only for ecological studies. Individual-level studies would more commonly use some measure of exposure and dose rather than ambient UV irradiance. UVI does not quantify the impact of UV radiation on human skin – this requires a measure of exposure as well as irradiance, and probably for impact, a measure of skin type. The sentence needs to be rephrased – perhaps to note that UVI is the cornerstone of public health messages for sun protection, as a measure of ambient UV irradiance.*

Both sentences have been rephrased.

2.2 Model Validation

1. *Page 7-8: this sentence needs to be revised: "UVIOMI tends to be a lightly higher" (slightly?)*

Corrected.

2. *The comparison between the different UVI measurements is a little confusing – there are the modelled UVI values, the observed from ground-based spectroradiometers, and computed results based on OMI data. At the bottom of page 7 – UVIMean and UVIMedian are the values modelled from the CCMI using TUV; the 'observed climatological UVI' – is the OMI-based estimates? And then 'observations' refers to the measured ground-level data. It would be clearer to consistently use the abbreviations that*

*are used in Table 3, i.e. UVIOMI, UVIMean,UVIMedian and UVIGB and use those to refer to the different datasets. Also later – simulated, CCMI models, UVIMean – all refer to the same data.*

Specific abbreviations have been added in order to clarify the discussion.

> 3. *Page 8 "The global relative difference between these two data sets.." – it is not clear without looking at Fig 3 which two datasets are under consideration.*

These two datasets have been specified in the text.

2.3 Conclusion

> 1. *The final part of the conclusion focuses on possible impacts. For human health, the main driver of health risks is sun exposure behaviour. It is not just a direct association between UVI and health risk – this needs to be noted.*

We agree with the referee, this has been noted in the conclusion.

> 2. *As noted in the text, the results differ from those of previous assessments. Much weight is placed on these predictions, in terms of, for example, future health planning. Yet, when the estimates are so different, it is difficult to have any confidence in the findings from one model over another. The authors might address this consideration.*

This consideration has been addressed in the conclusion.

**Supplement:**

[revised manuscript text omitted]

---

## Author Comment (AC2) · 17 Sep 2018

**Response to the Referee's Comments**

Kévin LAMY

September 17, 2018

We would like to thank the referee for his thorough review. The comments have been very beneficial. We hope that the newer version has improved the paper.

**1 General comment**

*The influence of ozone, aerosols, surface reflectivity and clouds on UV in the past, present, and future were discussed in the 2015 Assessment of the effects of ozone depletion and climate change panel of UNEP (Bais et al., 2015) but this publication is not discussed. Bais, A. F., R. L. McKenzie, G. Bernhard, P. J. Aucamp, M. Ilyas, S. Madronich, and K. Tourpali (2015), Ozone depletion and climate change: impacts on UV radiation, Photochem. Photobiol. Sci., 14(1), 19-52. Results of that study are more recent and more relevant compared to those of Hegglin and Shepherd, 2009 and Bais et al., 2011, and should be discussed against the results of this study.*

*I have some doubts on using mean and median total ozone from all models to derive the UVI projections through the 21st century. By doing*

*this the inter model variability of the projected ozone is lost, and taking into account that UVI is not linearly related to ozone, the projected mean UVI should not equal the mean of the individual projections of UVI. The authors justify their decision to use the mean ozone on the results of Figure 3. However this figure shows annually averaged differences and does not provide any information on the spread of the seasonal differences as well as on the spread of the UVI projections by individual models. These two will determine the uncertainty of the projections and it would be essential to know how large thee uncertainties are. Unfortunately, if these assumptions are proven incorrect (or if they result into large uncertainties), then the whole discussion on the future evolution of the UVI projections will be questionable.*

We fully agree that it would be very interesting to add informations about the seasonal variations and the spread of the model. However, given the results of the Figure 3, even if there is a seasonal variation and spread between models, we believe that these variations are relatively low and that the median information is of interest. Moreover, making the whole treatment for every model and every RCP and very sensitivity test requires a high computational cost that we unfortuntaley could not afford.

**2 Specific comments**

1. *P3, L21: Also in Bais et al. 2015*

Corrected.

2. *P4, L1: The 11-year solar cycle affects UV (especially UV-B) through changes in strato-spheric ozone, while the direct influence is almost negligible.*

As suggested by the referee, precision has been added on this point.

> 3. *P5, L6: It would be very informative to show in Table 1, or in a separate table, which of the input parameters are provided by each model and which models have participated in the different experiments used in this study*

Except for aerosols, most of the input parameters are provided by the models cited in Table 2, this is why we choose not to detail this information. Models which have participated in the different experiments are specified in Table 2.

> 4. *P6, L4: For which spectral region was the surface albedo provided by the models? Usually the models provide the broadband albedo which is much different than the UV albedo. Please explain how this was handled.*

We used the broadband albedo provided by the models, it is indeed different than the UV albedo. CCMI output were already made and additional diagnostic output were not possible.

> 5. *P6, L7: I don't think that taking the median of three numbers is representative for the most likely value of the parameter. Probably in this case the mean optical depth would be more representative.*

We agree that the mean value may be more representative in general but we took the median to avoid eventually local erroneous values. This comment has been added in the manuscript.

6. *P6, L16-17: The two references refer to absorption cross sections of ozone and not to solar spectra. Assuming that this is a typo, using in the RTM calculations absorption cross sections different than those used for the retrieval of total ozone (Paur and Bass 1985 for GB instruments) could introduce inconsistency in the results.*

Corrected.

7. *P6, L21: Please state which is this simplification, no matter if it is discussed in another section.*

Corrected.

8. *P6, L22: Could you provide an estimate of how large would be the effect on the UV calculations due to using zonally averaged profiles?*

The distribution of ozone is mainly zonal, and in particular the altitude of the maximum concentration or the maximum concentration has a zonal distribution. On the other hand, the vertical distribution of ozone (such as that of aerosols) has a very small effect on UVR (compared to other important parameters influencing UVR variability such as total ozone or AOD). The use of a zonal mean therefore introduces only a minor effect on UV calculations. It is reasonable to neglect it compared to the other uncertainties associated with the method.

9. *P6, L27-31: Which models are used in the experiments? This is related to my previous comment (P6, L6).*

The models used for each experiments are listed in Table 2.

10. *P7, L7: In the comparison with GB data, has the altitude of each station been taken into consideration in the TUV calculations? UVI at Mauna Loa is by far greater compared to UVI at the seas surface for the same latitude.*

TUV calculations are made at sea level. We thought that the altitude of theses stations was relatively small (between 8m for Barrows and 370m for Lauder). We made an error due to the NDACC file (.mku file) from the Mauna Loa station which report a 3m altitude station. It is apparently about 3km above sea level. This should explained part of the discrepancy between our modelling results and the stations measurements. A remark has been added in the manuscript on this subject.

11. *P7, L15: In the monthly climatology, were data of the 15th used only (as with satellite data) or the mean of the entire month? How missing data were handled?*

For the monthly climatology we used a 10 day average around the 15th. It is now specified in the manuscript.

12. *P7, L29-31: It is not clear which the satellite data are used. Is always the 10 day average around the 15th is used, or only when data on the 15th are missing? How the clear-sky satellite data are selected?*

It is always the 10 day average satellite data which are used. It is now specified in the manuscript.

13. *P7, L31: As it appears from the text, for the monthly GB climatology all available data were used, but for the satellite day only the measurement*

*on the 15th (or the 10- day average). If this is true, the data used in Figure 1 are inconsistent and I do not understand how one can compare these datasets with the model results.*

A 10 day average is used in both ground based and satellite data selection. It is now specified in the manuscript.

14. *P7, L32- P8, L10: Please check this section and make the discussion clearer. I suggest focusing the discussion on differences between model and GB data and not on differ- ences between GB and OMI because this is not the main subject of this paper. The changes reported in the abstract and the conclusions (-4 to 11all in this section. Furthermore it would be good to report in Table 3 the spread of the model differences to GB for each station (e.g. the standard deviation).*

15. *P8, L22: Averaging the UV index over the entire globe is not a good approach due to large latitudinal (and seasonal) differences.*

For this computation we computed the difference between months and then took the average. Nonetheless we fully agree that it is not a good approach due to latitudinal and seasonal differences. But in this case, we aim at having a first idea of the global difference of behaviour between the models. This part of the article is not intended to study the difference between the model+TUV and OMI outputs in detail, but rather to estimate the overall behaviour of the models towards OMI measurements in order to infer their homogeneity. The result, presented here, even global, is consistent with the publications comparing OMI and ground stations.

Specifications on the averaging process and on the limitation of this sensitivity analyses have been added.

16. *P8, L31-34: The results presented in Figure 3 are yearly averaged. However, the vari- ability on predicted ozone varies seasonally (see Dhomse et al., 2018) and differences in UVI may also have a seasonal effect, which now is suppressed. Moreover, the test is performed for a period (2000-2010) when inter-model ozone variations are smaller than later in the 21st century. As changes in the UVI are nonlinearly related to ozone, the assumption that total ozone evolution in the 21st century can be represented by the mean or median is not a safe choice. I suggest checking whether the seasonal behavior of the differences is still within the reported limits, particularly in the southern high latitudes.*

We fully agree with the referee on this point. We added a remark about theses concerns in the manuscript. It is not proposed here that the evolution of TOZ be represented by the mean or a median, but it is just suggested that averaging TOZ before or after the use of TUV does not make a significant difference, which allows a significant saving in computing time. It is true that a significant seasonal variability of total ozone coupled with the non-linearity of the relationship between UV and ozone (low non-linearity) could induce a bias, but this would only be problematic at high latitudes in the southern hemisphere, over a relatively short period. We can see that the most important differences on fig 3 are seen in this region of the globe. But this should not call into question the general method.

Unfortunately, for now we do not have the resources to investigate longitudinal and seasonal changes but it should be thoroughly analysed on the future study from AerChemMIP results along with detailed AOD effects.

17. *P9, L4: Moreover, Figure 3 shows the sensitivity due to averaging of only the total ozone. What is the uncertainty introduced by the averaging of the other input param- eters; the ozone and temperature profile, the surface reflectivity, the optical depth of aerosols?*

The referee is right to bring up this matter. Nonetheless, as Bais et al. (2015) noted, absorption by ozone is the dominant factor controlling levels of surface UV for clear skies and low aerosols conditions. Our study is made exclusively in clear sky conditions and for aerosols AerChemMIP results would be a better context to investigate the uncertainty associated with aerosols along with profile and surface reflectivity differences.

18. *P10, L17-28: The changes derived in (Bais et al., 2011) have been based on a different reference period (as noted in the text); therefore the results of this study are not directly comparable, particularly as in these early years ozone variations were quite large.*

We kept the discussion on Bais et al. 2011 results, while it is not fully comparable, Bais et al. (2011) study and ours are similar and should be discussed.

19. *P12, L31-25: Effects of AOD on UVI have large longitudinal variability (see Bais et al., 2105) which is suppressed when taking zonal averages. It would be interesting to compare the effects of TOZ and AOD on UVI of this study with those reported in Bais et al., 2015.*

A discussion has been added on this subject.

20. *P13, L4: As mentioned above AOD exhibits large spatial variability, therefore the results found for zonally averaged UVI changes should not be generalized for all latitudes. For example, AOD over China will decrease substantially by 2100 but at similar latitudes over the Pacific the effects are almost negligible.*

We added a sentence at the beginning of the analysis emphasizing that the results are in zonal average and should not be generalized. Along with the discussion on Bais et al (2015) results and ours, concern over this point was also expressed.

21. *P13, L24: Please discuss to which level of accuracy the UVI can be reproduced.*

This was discussed P7, L24,28. Specific values from Brogniez et al. (2016) has been added in P7 and P13.

22. *P14, L1: State here that UVI projections are for cloud-free skies. Cloud effects can alter significantly the predicted changes at high latitudes.*

Corrected.

23. *P14, L6: State the period for the increase.*

Corrected.

**3 Technical comments**

24. *P4, L14: Replace "increment" with "increase"*

Corrected.

25. *P6, L26-27: Replace to: ". . . with each other, we defined two experiments from two sets of models. These are summarized . . ."*

Corrected.

     26. *P8, L11: I suggest replacing "wind variability" with "stratospheric circulation"*

Corrected.

---

## Author Response (AR2)

**Response to the Referee's Comments 1**

December 30, 2018

**1 General comment**

*The authors have responded to my queries, but in some cases the response is not quite correct and needs further revision (see below). The paper is difficult to follow  in many parts the language is in poor English, there are some very long sentences, and some sentences that seem to be out of place. The simulations need to be better described I could not find anywhere what the various bits of senC2rep26 stood for  26, presumably RCP 2.6. The simulations need to be listed and explained in one of the tables. UV is inconsistently used as UV, UVR, UV radiation  it should be consistent. Also, on page 6 TSO2 is defined then, TSO2 used.*

We would like to thank the referee for his/her thorough review. The comments have been very beneficial. We hope that the new version has improved the paper. With the help of the co-authors, an effort has been made on improving the language. In order to better describe the different simulations, a new table has been added (Table 3). UV is now only used as an acronym for ultraviolet. For example, ultraviolet radiation is now written as "UV radiation".

**2 Specific comments**

**2.1 Introduction**

*1. 1. Page 3, Line 8.  Sentence starting Overexposure should be a new para. This section on health risks is not quite right  and the detail given about the exposures is not necessary (and due to sunburns during childhood years is not correct  there is an increased risk associated with sunburns during childhood years, but sunburn and higher sun exposure at all ages are important to melanoma risk). I would suggest replacing lines 8-14 with the following: Exposure to UV radiation has both adverse and beneficial effects on human health. Overexposure increases the risk of skin cancers, e.g. cutaneous malignant melanoma and keratinocyte cancers, and a range of eye diseases. Underexpsoure increases the risk of vitamin D deficiency; vitamin D is critical to healthy bones. It is common in health research and public health messaging to use the UV Index (UVI) as a measure of erythemally (sunburn) weighted UV irradiance.*

We agree with the referee, his suggestion has been implemented. P3, L6

*2. Page 3. The sentence starting Dhomse et al on line 22 seems to be an add-on that is not in the correct place.*

Corrected.

*3. Page 3, Line 33: why is it Consequently  this seems to be the wrong word, since the rest of the sentence does not seem to relate to the sentences that precede it.*

Corrected, it is now replaced with "In addition". P3, L29.

4. *Page 4, Line 1  stratospheric ozone absorbs UV-B radiation (not UV)*

Corrected, P3, L32.

5. *Page 4. Typographical error line 6  scaterring*

Corrected. P4, L3.

6. *Page 4. Line 19  this is a very long sentence with a large number of acronyms, it is very difficult to read.*

The sentence has been cut in half. Nonetheless, we kept the numerous acronyms in order to correctly describe and identify the different partners of the CCMI project.

7. *Page 5: the information about RCPs could probably go in the Introduction.*

The information about RCPs could go in the introduction but we preferred to kept them in this section which also introduce the A1 scenario and the CCMI simulations. (Section 2 and Table 3 2)

**2.2   Data and Methodology**

1. *The A1 scenario is not explained*

Informations about the A1 scenario has been provided in the newly added Table 3.

2. *The simulations are not explained (see general comments). I dont think it is enough to say they are described in a different paper. In addition, this last paragraph on page 5 seems very disordered. It is difficult to follow.*

Corrected. A new table (3) summarized the simulations and scenarios. It is also referred in the manuscript (P6,L5).

3. *Page 6: line 5-7. This is already in the list on page 5. Could be deleted in one location.*

Corrected, the fullname were deleted, only a short list remain (P6, L6).

4. *Page 6, line 28  two sensitivity simulations  is this C2fODS & C2Fghg? If you call these sensitivity simulations at the first mention, this will be clearer to the reader.*

Corrected. (P6, L1)

**2.3   4.3 Effects of greenhouse gases**

1. *Page 12: these are simulations based on refC2 this has already been noted earlier and should be deleted here.*

Corrected.

2. *Page 12, last line. Change decent increase to a moderate increase  something more scientific than decent*

Corrected. (P13, L19)

**2.4  4.4 Other effects affecting UVI**

1. *Page 13. Sentence at line 18-19 seems to be out of place.*

We are not sure which sentence the referee are refering. If it is referring to this passage: "Both TOZ and AOD drive [...] strong effect on UVI", we do not think that the sentence are out of place.

**2.5  Conclusion**

1. *This seems quite well written and provides a nice summary of the results although it is more a summary of the results than a conclusion.*

Corrected. This section is now renamed as "Summary".

2. *Page 15, line 7-8. Not clear what this sentence means: This study focused on UVI which does not directly impact human health. UVI is erythemally weighted UV irradiance  so is weighted for a human health effect.  The following sentence about further investigation is very nave and should be deleted  behaviour is a very important moderator of any link between UV irradiance and human health impact.  It would be impossible just to change UV irradiance and look for human health impacts, without taking a lot of other parameters into account. Maybe best just to say it is complicated by human behaviour and not recommend further investigations.*

The reviewer is correct, the sentence was not very clear. The reviewer's suggestion was implemented. (P15, L29-31)

**2.6  Tables**

1. *Table 2 needs a much better description of what it is showing.*

Corrected, more details have been added in the caption of this table.

**Response to the Referee's Comments 2**

December 30, 2018

**1 General comment**

*The revised manuscript has been improved but there are still weaknesses that must be taken into consideration before it is accepted for publication. See specific comments below. The language of the manuscript needs further attention, as well as the flow of the text in some cases. Apart from the corrections suggested in the first review, there has been no attempt to improve it. I am sure that one of the co-authors could help in this context.*

We would like to thank the referee for his/her constructive comments. The comments have been very beneficial, we hope that this revised manuscript will answer the referee's interrogation. As suggested by the reviewer, the language has been improved with the help of the co-authors and some paragraph have also been reworked in order to improve the flow of the text.

**2 Specific comments**

1. *(General Comment). I still believe that the use of average ozone columns in RTD simulations does not allow assessing the uncertainty of the presented UVI simulations. I understand however, that repeating the calculations is not possible at this stage of the study.*

We thanks the referee for his understanding of the limitations of our study. We fully agree that a UVI computed for each ozone columns from each models would give us a better understanding of the uncertainty associated with the UVI simulations. It is a perspective that could be applied for future studies.

2. *Comment (4): P6, L6: The authors response in my comment on the spectral characteristics of the albedo used in the UV calculations has not been reflected in the text. Please add.*

It is now reflected in the text. P6, L7.

3. *Comment (8): P6, L24: Another case of responding to a comment without passing the explanation in the text. Please explain for the sake of the readers, why using zonally averaged ozone and temperature profiles does not introduce important uncertainties.*

It is now explained in the text. P6, L26-30.

4. *Comment (14) was not taken into account, nor was a response from the authors. See below: P7, L32- P8, L10: Please check this section and make the discussion clearer. I suggest focusing the discussion on differences between model and GB data and not on differences between GB and OMI because this is not the main subject of this paper. The changes reported in the abstract and the conclusions (-4 to 11%) are not discussed at all in this section. Furthermore it would be good to report in Table 3 the spread of the model differences to GB for each station (e.g. the standard deviation).*

Comment (14) was indeed not addressed in the previous review. We are sorry for this oversight. As suggested by the reviewer, the discussion is now centred on the difference between modelled UVI and GB. The changes reported in the abstract are now coherent with the changes discussed in this section. Spread of the model differences to GB for each station is now reported in Table 4.

> 5.   *I havent identified in the text any Specifications on the averaging process and on the limitation of this sensitivity analyses have been added., as quoted by the authors.*

Specifications on the averaging process has been added in the caption below Figure 2. Limitation of this sensitivity analyses have now been added in the text (P9, L11-16 ).

> 6.   *Comment (18): I insist on my previous comment that you cannot compare changes between two studies that are based on different periods, without mentioning that explicitly in the text. If you see Figure 3 of the Dhomse 2018 paper, it is obvious that total ozone was much higher in 1960 than in 1980 almost everywhere except in the tropics. Taking these differences into consideration you could estimate the additional change in UVI that is induced by the ozone difference. For example, in SH midlatitudes ozone is about 2.5% higher in 1960 meaning that UVI would be at least 3% lower in 1960 than in 1980. This change is comparable to the changes that Bais 2011 and the present study have reported.*

We agree with the reviewer on this point. Nonetheless, instead of estimating UVI additional changes due to ozone difference between 1960 and 1980 from (Dhomse, 2018) paper and by using the radiative amplification factor. (I suppose this is how you proposed 3% UVI change from 2.5% TOZ change, for a radiative amplification factor of about 1.2?). We choose here to simply estimate graphically UVI changes between 2100 and 1960 from Figure 2 of Bais et al (2011) paper. Both values are now reported in the Table 5 of our study; Computed values by Bais et al (2011) (Table 2) for the difference between 2100s and 1975s and estimated value from Bais et al (2011) (Figure 2) for the difference between 2100 and 1960.

> 7.   *Comment (21): The first sentence of the conclusions should indicate how good the UVI is reproduced by the model. Here is a suggestion: We have shown that the use of CCMI model data with a radiative transfer model (TUV) can reproduce the current climatological values of clear-sky UVI derived from measurements, in most cases, to within 5%. This is my take on message from the results of Table 3.*

We agree with the reviewer, his suggestion is now placed at the beginning of the conclusion. (P14, L27)

**3   New comments**

> 1.   *P4, L8: (Mayer et al., 1998) is not a good reference for cloud enhancement. This paper discusses the apparent large increases of total ozone under optically thick clouds. Possible alternatives:*

Both alternatives references for cloud enhancement are now used in the manuscript. (P4, L6)

> 1.   *P8, L9-10: It is now clear what you mean with Here, a similar conclusion can be drawn,. What is the compulsion and to what it refers? The previous conclusion was about OMI being higher than GB observations. Please revise the text. Moreover, an overestimation of 10% was found for Palmer but there is not discussion on what could cause this discrepancy.*

A discussion has been added for Palmer. This part of manuscript have been reworked. (P8, L11-33)

1. *P8, L15: The altitude effect of UVI is the order of 7-8% increase per km altitude (McKenzie et al., 2001). This means that, for Mauna Loa, the relative difference between modeled for sea-level and measured UVI should be at least -25% while in Table 3 the difference is about 5% and of opposite sign. In Figure 1 both the median and mean based model results are higher than the measurements with one exception in June where the difference is in the right direction but much smaller than it should be. Therefore, the statement in line 17: Nonetheless, at low and mid-latitudes the UVI differences observed are fully compatible... , is invalid, and of course the comparison is not good for unidentified reasons.*

We agree with the reviewer, the comparison is not good for Mauna Loa for unidentified reasons. Clear-sky filtering could be responsible for the discrepancy. Mauna Loa is located on a tropical island, where cloud cover vary rapidly. At the other tropical station (Saint-Denis), the clear-sky filtering was investigated and validated in Lamy et al. (2018). For Mauna Loa, we used the data as provided by NDACC. This has been developed in the manuscript. (P8, L26-33)

1. *P9, L5: The newly added sentence refers to latitudinal differences, but these are already taken into account in Figure 3. Consider revising to: Due to seasonal and inter-model differences, this approach has limitations*

Corrected (P9, L26)

1. *P14, L11-12: These sentences belong to the introduction (already referenced). They are not conclusions of this paper, and as they are written create some confusion as to what means reproduced UVI.*

The sentence has been reworked. It is still in this section which is now called Summary.

1. *P14, L23: According to Fig. 4, UVI does not return to 1960 levels at high southern latitudes when TOZ returns. It stays 5-8% above except for RCP 8.5 when TOZ exceed the 1960 levels. Please rephrase.*

Corrected P15,L8-10

1. *P14, L27: The statement: In mid-latitudes, TOZ should increase between 1960 and 2100 in both hemispheres for all RCPs except RCP 8.5., is not correct: a) TOZ must be replaced with UVI. b) UVI increases between 1980 (not 1960) and 2100 only in the NH midlatitudes. In the SH midlatitudes the pattern is completely different, as it appears from Figure.*

Corrected. This sentence now address correctly UVI differences between 2100 and 1960 in the mid-latitudes, this sentence refers explicitly to Table 5. (P15,L13) For the SH midlatitudes, you mentioned a different pattern observed from Figure (not numbered) , i suppose it's referring to Figure 5. From this figure, there is indeed a different pattern of changes depending on the scenarios. This is now mentioned in the manuscript (P15, L15)

1. *P15, L3-5: Please state to which time frame refers the quoted variability 0-3%. Surprisingly, in the next sentence the variability reaches 10%, without saying to which latitude or period it refers. Please clarify.*

Precision has been added on the time frame and figure. The variability reaches 10% only during the summer months, as demonstrated by Figure 5, this is now specified in the manuscript. P15 L25-27.

1. *Table 3 caption: In the last line it is stated For Barrow and Palmer station we selected the six months of their respective summer. Since summer has only 3 months. It is better to explicitly name the months that were used.*

This precision has been added in the corresponding Table (now Table 4).

**4  Technical**

1.  *P2, L12: I suggest rephrasing to: Higher increases in UV index are projected*

Corrected, P2, L12.

2.  *P2, L19: I suggest rephrasing to: of ozone to 1960 values, with a corresponding pattern of change observed on UVI,*

Corrected. P2,L19.

3.  *P7, L7-8: As there is no other second-level heading, heading 3.1 Model Validation should be deleted. I suggest to stop heading 3 at the colon to become: 3 Model Validation.*

Corrected. P7, L12.

4.  *P14, L14: Replace "ranges" with "ranging"*

Corrected, P14, L27.

5.  *P15, L4: The word "Logically" does not make sense. Please remove. Also, replace hemisphere; with hemispheres*

Corrected. P15, L264.

6.  *Figure 4: I suggest to maintain consistency in the vertical axes labels; some of them are drawn with one decimal, some without. Include a note in the caption, that the scale of the vertical axes is not the same for all panels. Same for Figure 6.*

Figure 4 and 6 have been remade with attention to the label of the vertical axe. A comment has been added in the caption regarding the different vertical scales used.

7.  *Caption Figure 6: The last sentence has been copied form Figure 4. But this figure does not show the four RCPs, but the three experiments. Please correct.*

Corrected.

8.  *Caption Figure 8: The word "modelized" does not exist in English. Please replace with modelled (three occurrences). Please change also from 2000-2010 values to from 2000 to 2010. Same for Figure 9.*

Corrected.

9.  *Axes titles in Figures 4,6,8,9: Remove the word "percent" since there is already a [%] sign.*

The word "percent" has been removed from Figures 4,6,8 and 9.

[revised manuscript text omitted]

---

## Author Response (AR3)

**Response to the Editor's Comments**

May 17, 2019

*Co-Editor Decision: Reconsider after major revisions (22 Feb 2019) by Paul Young*
*Comments to the Author: There are still major issues with the manuscript as highlighted by the reviewers. I believe that there are some results of potential interest to the community here, but I would ask that you particularly consider:*

*1. The Mauna Loa data used in the model evaluation, as highlighted by Reviewer #2.*

*2. The language and structure of the paper, as mentioned by both reviewers. There are still several issues with these, which means that would-be readers might just not persevere with the manuscript.*

We would like to thank the editor for their interest in our work. The comments have been very beneficial. We hope that the new version has improved the paper. Regarding the specific comments:

1. As highlighted by Reviewer #2, there was an issue with Mauna Loa data, specifically with the clear-sky flag. We contacted the P.I of the instrument and obtained an updated clear-sky filter. The consequent changes are detailed in the answer to Reviewer #2.

2. The language and structure of the paper were reviewed. We have restructured section 4.1 and 4.4 and added subsections in order to clarify the reading. The results for each region (high, mid-latitudes and tropics) are now presented in their respective subsection. In others section, notable section 2 and 3.1, paragraphs have been also been reorganized to clarify the methodology or the result. The language of the paper has also been checked thoroughly.

**Response to the Referee's Comments 1**

**May 17, 2019**

We would like to thank the referee for their interest in our work. The comments have been very beneficial. We hope that the new version has improved the paper.

We have checked the general and specific comments and have made the consequent changes. Responses to the reviewer and changes in the revised manuscript are as follows.

**1    General comment**

> *Some issues in the paper have been fixed, but there remain a number of issues with incorrect and inconsistent language and punctuation (including some that the author response would suggest had been fixed). Poor structure of some of the main results paragraphs make for very confusing reading.*

We have reviewed the entire manuscript carefully.

We have restructured section 4.1 and 4.4 and added subsection in order to clarify the reading. The results for each region (high, mid-latitudes and tropics) are now presented in their respective subsection. In others section, notable section 2 and 3.1, paragraphs have been also been reorganized to clarify the methodology or the result. Language and punctuation have been reviewed as well.

**2    Specific comments**

1. *Page 3 Line 2: specify "stratospheric ozone layer"*

Corrected. (Page 3, Line 2)

2. *Page 3, Line 13: UV radiation also impacts the biosphere (missing word)*

Corrected. (Page 3, Line 13)

3. *Page 3, Line 23: future surface UV radiation*

Corrected. (Page3, Line 18)

4. *Page 3, line 33: affecting surface levels of UV radiation (also line 34 and a number of other places in the manuscript  please find and replace where necessary)*

5. *Page 4, line 2  should this be larger SZA rather than higher?*

Corrected. (Page 4, Line 3)

6. *Page 4, line 13  CCMI not yet defined (is defined further down in line 18)*

CCMI and CCMVal-2 are now defined here (Page 4, Line 15) and abbreviated after.

7. *Page 4, line 29  bracket missing before (Meinshausen et al)*

Corrected. (Page 4, Line 35)

> 8. *Page 5 heading  should this be Modelling UV irradiance?*

Corrected. (Page 5)

> 9. *Page 5, line 5  is integrated the correct word? UVI is also irradiance, so it is not clear what integration is performed to get from irradiance to UVI*

Corrected, it is now stated that the "resulting weighted solar spectral irradiance is integrated". (Page 5, Line 10)

> 10. *Page 6, line 4: summary rather than summaries*

Corrected. (Page 6, Line 9)

> 11. *Page 6, line 13: rephrase eventually local erroneous values*

Replaced by "possible local erroneous values". (Page 6, Line 23)

> 12. *Page 8: line 27: missing capitals on Sept and Dec*

Corrected. (Page 9, Line 3)

> 13. *Inconsistent capitalisation of Northern and Southern Hemisphere*

Corrected. We followed ACP guidelines for capitalization. Hemisphere is a proper noun therefore Northern and Southern Hemisphere are now capitalized. "southern/northern high latitudes, mid-latitudes or tropics" are not capitalized.

> 14. *Page 9, line 13: suggest add While we recognise that this is not the best approach*

Corrected. (Page 9, Line 31)

> 15. *Page 9, line 18: delete one*

Corrected. (Page 10, Line 2-3)

> 16. *Page 9, line 30  this has already been said on page 6, line 24. Remove the repetition.*

Corrected. (Page 10, Line 16)

> 17. *Page 10, line 33: In the Northern Hemisphere  which latitude band is this referring to?*

Corrected. (Page 11, Line 21)

> 18. *Page 10-11  last para page 10 and continued as first para page 11.  This paragraph jumps back and forth between north and south latitudes, and between different latitude bands  this makes it very hard to follow.  It should be revised with a better structure.  (the same lack of structure is apparent in the next two paragraphs, and needs revision).*

Corrected, the entire section 4.1 have been reorganized and rewritten, this paragraph correspond to section 4.1.1 (Page 11).

> 19. *Page 14, line 25 climatological  spelt incorrectly.*

Corrected (Page 16, Line 6).

20. *Page 14, line 31. Sentence starting While the mean is not a sentence needs to be rephrased.*

Corrected (Page 16, Line 13).

21. *Page 15, para starting on line 12. This is very confusing: UVI changes in this region are not homogeneous; (not clear what region is meant, but perhaps mid-latitudes as at the beginning of the para). But next part of the sentence discusses southern hemisphere 20-30 - which by the definition in the paper is low latitude/tropics.*

This paragraph has been rewritten. (It starts on page 16, line 27.)

22. *Titles for tables 2 and 3 are not in the correct order*

Corrected.

23. *Relative difference definition is this 100*? (under several tables and figures). Below Figure 3, RD for UVI is defined as 200* - is this correct?*

200 is the correct value. In the other figures, the difference were computed against previous values (past values) or reference values (ground based stations or satellites). Therefore the denominator used to compute the relative difference is this reference value. Here, there is not clear reference between $UVI_{MEAN}$ or $UVI_{ALLM}$. We choose instead to use the mean of both values as the denominator which raise a factor 2.

**Response to the Referee's Comments 2**

**May 17, 2019**

We would like to thank the referee for their interest in our work. The comments have been very beneficial. We hope that the new version has improved the paper.

We have checked the general and specific comments and have made the consequent changes. Responses to the reviewer and changes in the revised manuscript are as follows.

**1  General comment**

> *The paper has certainly been improved. My few comments below (except the second) are mainly minor aiming at improving the readability of the paper. As long as the 2nd comment is taken into account I would have no objections to recommend publication of the manuscript.*

We appreciate the positive feedback from the reviewer. Concerning the second comment we have analysed the issue thoroughly, our answers and changes made in the manuscript are detailed below.

**2  Specific comments**

> *1. As cloud effects in UV radiation are not considered, I suggest changing the title to: "Clear-sky ultraviolet radiation modelling using output from the Chemistry Climate Model Initiative"*

Accepted.

> *2. In the model validation (section 3) I have still some concerns about the comparisons for Mauna Loa. The authors try to explain the differences of about 59% between modeled and measured values of the UVI, but they do not realize that they should try to explain why the differences are not -25%. As explained in my review (3rd new comment), due to the altitude of the site, the measured UVI should be at least 25% higher than the modeled value that refers to sea-level. In Table 4 the modeled value is higher by 5%, which makes a difference or about 30%. I think that something is wrong with the data used for Mauna Loa. A recent poster presentation by the data providers shows that the noon UVI should be between 15 and 16 while in Figure 1 the data are between 11 and 14. Similarly the minimum values should be about 8 and not 5-6 as in Figure 1. I suggest checking the data that were used. It might be that the averages are from all data and not only cloud-free as reported in the paper. If this is the case, then the comparisons are invalid and this should be explicitly stated in the paper.*

The reviewer were right to be concerned about the comparisons for Mauna Loa. After investigation we found that the clear-sky filter provided by the station data were not coherent. Therefore, we contacted the team in charge of Mauna Loa data. They provided us with an updated clear-sky filter. The noon UVI measured at Mauna Loa (Figure 1) ranges between 14 and 15 UVI unit

which is coherent with the poster provided by the reviewer. Similarly, the minimum values are about 8 UVI unit.

Consequently, measured values of clear-sky UVI (made at about 3km above sea level) are now correctly higher than modelled UVI (modelled at sea level). During the summer month of July and August, measured UVI are about 25% higher than modelled UVI, which is coherent with what the reviewer pointed out. The mean relative difference for all months is about 12-13%.

We would like to thanks B. Liley, R. McKenzie and M. Kotkamp for their help on this matter. We also would like to thank the reviewer for pointed out this matter and helping us understanding the issue.

> 3. *I havent seen any acknowledgments to the data provides of the stations used in this study. As, I am not fully aware of the data policy of NDACC, I am just warning the authors to check if there is any special requests for this.*

The acknowledgements were already provided (Page 31), we updated them by citing the P.I of the different station.

**3 Grammatical comments**

> *I have spotted the following typographical and grammatical defects. I mainly provide the correct form of the words, unless otherwise specified.*

> *P6, L4: summary*

Corrected (P6, L9)

> *P7, L19: by spectroradiometers*

Corrected (P7, L29)

> *P7, L19: from the spectral irradiance measurements.*

Corrected (P7, L29)

> *P8, L16-17: they range (2 occasions)*

Corrected (P8, L30-31)

> *P8, L23: UVI units*

Corrected (P9, L10)

> *P8, L27: Capitalize September December*

Corrected. (P9, L3)

> *P8, L31: stations*

Corrected, the corresponding sentence has been removed.

> *P8, L32: varies, patterns*

Corrected, the corresponding sentence has been removed.

> *P8, L34: the differences between Mauna Loa and Palmer stations, delete a (part of)*

Corrected. (P9, L14)

*P9, L6: change estimate to verify*

Corrected. (P9, L21)

*P10, L16: drive*

Corrected. (P15, L3)

*P10, L24: this will be verified to as discussed*

Corrected. (P11, L10)

*P10, L30: changes*

Corrected. (P11, L19)

*P11, L1: UVI is almost constant, due to effects of AOD changes (see section 4.4).*

Corrected. (P11, L24)

*P13, L22: we found an UVI we discussed UVI*

Corrected. (P11, L24)

*P13, L29: 6% change in UVI in 2100*

Corrected. (P15, L10)

*P14, L5: clear-sky UV levels*

Corrected, the corresponding sentence has been removed.

*P14, L6: Delete percent (2 occasions)*

Corrected. (P15, L6)

*P15, L2: account for to explain*

Corrected. (P16, L18)

*P15, L6: cloud effects*

Corrected. (P16, L22)

*P15, L8: TOZ returns*

Corrected. (P16, L25)

*P15, L10: will arrive to will occur*

Corrected. (P16, L26)

[revised manuscript text omitted]

---

## Author Response (AR4)

**Response to the Reviews**

July 4, 2019

**1 Response to the Referee's Report 1**

*The paper has been revised substantially and previous comments have been addressed properly. Please consider the following technical corrections (page/line).*

We would like to thank the referee for their interest in our work. The comments have been very beneficial. As suggested by the referee, the following corrections have been made.

*9/13: I suggest replacing the sentence: Both modelled UVI and UVIOMI are not able to reproduce ground-based measurements of UVI for this period. with: Both modelled UVI and UVIOMI are biased low with respect to the ground-based measurements of UVI for this period, due to the altitude difference of the monitoring site.*

Corrected.

*9/16: replace explained with explain*

Corrected.

*14/23: replace effects with factors*

Corrected.

*15/5: It sounds unreasonable that 80% change in AOD results on only 2% Change in UVI. I suggest adding after the sentence  With fixed TOZ (EXP3FAOD), the 80% decrease of AOD result in a 2% increase of UVI. This small effect is due to the small absolute values of the zonally averaged AOD.*

Corrected.

**2 Response to the Editor**

*Many thanks for your updated manuscript and addressing the comments and concerns of the reviewers. I am happy to proceed to publication subject to technical corrections. In addition to the technical corrections suggested by reviewer 2, I would also ask the authors to consider their figures:*

We are grateful to the Editor for the valuable suggestions provided. As suggested by the Editor, the following modifications have been made.

*1. Please ensure that the colours in the line plots are appropriate for colour-blind people. For example, see the resources at `http://colorbrewer2.org/`*

Colours have been checked for colour-blind people. The figures were mostly made with NCL, therefore we used the colormap provided by NCL. (`https://www.ncl.ucar.edu/Document/Graphics/ColorTables/Aid_in_color_blindness_cat.shtml`)

*2. For Figure 6 in particular, consider how much more the axes are emphasised (through their thickness) than the data you are trying to show. Can this be improved?*

Axes and grid for Figure 6 and 4 have been shrinked.